# Hepatitis C virus NS3 helicase contributes to (−) strand RNA synthesis

Philipp Ralfs [1], Stéphane Bressanelli [2], Lina M. Günter [3], Alexander Gabel [3,4], Paul Rothhaar [1], Kyle J. Price[5], Thibault Tubiana [2], Mathias Munschauer [1,3,4], David N. Frick[5] & Volker Lohmann [1,6] ✉

Many positive strand RNA viruses encode helicases, but their distinct functions in viral replication cycles is poorly understood. Here, we identify a mutation in the helicase domain of HCV non-structural protein 3 (NS3h), D1467G, which specifically affects (−) strand synthesis, phenocopying mutations in the 3′ untranslated region of the genome. D1467G does not impair helicase activity in vitro or the binding of NS3h to critical cis-acting RNA elements, but reduces the interaction of NS3h and NS5B polymerase, potentially contributing to defective (−) strand synthesis. AlphaFold predictions of complexes between NS3h, RNA and/or NS5B suggest that NS3h both remodels the cis-acting RNA elements and unwinds the terminal stem-loop of the HCV genome rendering the template accessible for de novo initiation of (−) strand synthesis by NS5B. Overall, our study provides evidence for a defined function of a viral helicase in (−) strand genome synthesis of a positive strand RNA virus.

Hepatitis C virus (HCV) is an enveloped virus with a RNA genome of positive polarity and a member of the *Flaviviridae*[1]. Based on sequence diversity, Hepatitis C viruses are classified into eight genotypes (gt1–8)[1]. The 9.6 kb HCV RNA genome of positive polarity contains a structured 5′UTR, a single ORF encoding a polyprotein and a structured 3′UTR[2]. The 5′UTR contains an internal ribosomal entry site (IRES) enabling cap-independent translation[3]. Translation at the rough ER generates a single polyprotein of about 3000 amino acids, which is cleaved by viral and cellular proteases into 10 individual functional proteins[3]. HCV proteins can be divided into two categories: proteins essential for particle assembly and release (Core, E1, E2, NS2, p7) and proteins required for viral RNA replication (NS3, NS4A, NS4B, NS5A, NS5B)[4]. NS3/4A, harbouring protease and helicase activities, and NS5B, the viral RNA-dependent RNA polymerase (RdRP), are the main enzymes involved in protein processing and RNA replication. All HCV proteins are membrane proteins or membrane-associated, and their expression leads to the generation of the membranous web, consisting of double membrane vesicles (DMVs) in proximity to the ER[5] and representing the presumptive site of viral RNA replication. Transcription of (+) viral RNA into (−) RNA generates the double-stranded (ds) RNA replicative intermediate. The (−) strand is then used as a template for excess (+) strand synthesis. Newly synthesized (+) viral RNA is then either translated into viral proteins or packaged into viral particles[6].

RNA remodelling proteins possess various functions in cellular metabolism and viral RNA replication[7,8]. Helicases are RNA remodelling proteins possessing RNA or DNA unwinding activity, which directionally unwind nucleic acids (either 3′−5′ or 5′−3′). HCV Non-structural protein 3 (NS3) is a bifunctional enzymatic protein[9–11], consisting of a N-terminal serine protease and a C-terminal helicase. The serine protease requires NS4A as a cofactor and is involved in HCV polyprotein processing as well as the cleavage of cellular proteins[10]. The HCV C-terminal NS3 helicase (NS3h) contains a 3′−5′ helicase belonging to the DEXH family of helicase super family 2 (SF2). NS3h unwinds DNA and RNA in an ATP-dependent manner in vitro[12–15] and contributes to

[1]Department Infectious Diseases, Molecular Virology, Heidelberg University, Medical Faculty, Heidelberg, Germany. [2]Université Paris-Saclay, CEA, CNRS, Institute for Integrative Biology of the Cell (I2BC), Gif-sur-Yvette, France. [3]Helmholtz Institute for RNA-based Infection Research (HIRI), Helmholtz-Center for Infection Research (HZI), Würzburg, Germany. [4]Institute of Medical Virology, University Hospital Frankfurt, Goethe University, Frankfurt am Main, Germany. [5]University of Wisconsin, Milwaukee, WI, USA. [6]Present address: Heidelberg University, Medical Faculty, Department of Infectious Diseases, Molecular Virology, Section virus–host interactions, Centre for Integrative Infectious Disease Research (CIID), INF 344, Heidelberg, Germany. ✉e-mail: Volker.lohmann@med.uni-heidelberg.de

infectious particle assembly[16–24]. Despite NS3h being well-studied, its contribution to HCV replication remains incompletely understood, and it is unclear which RNA structures are unwound by NS3h. Potential candidate structures would be the double-stranded (ds)RNA replicative intermediate as well as highly structured RNA elements in the 5'UTR, within the coding region in NS4B and NS5B[25], as well as in the 3'UTR. The 3'UTR contains a variable region containing a polyU/C stretch of variable length as well as a highly conserved and structured 98 nt long 3'X region[26], the presumptive promoter of (−) strand synthesis. The presence of large, conserved stem loops and reports that 3'X is a poor NS5B template in vitro[27] make it a potential target for unwinding by the NS3 helicase, thereby enabling initiation of NS5B-mediated (−) strand synthesis. Since the NS3 helicase translocates in a 3'−5'-helicase direction, the 3' end of the X tail is the potential binding and initiation site.

The dsRNA replicative intermediate is a potent pathogen-associated molecular pattern (PAMP) sensed by pattern recognition receptors (PRRs) in the cytoplasm (RIG-I-like receptors (RLRs)) and the endosome (Toll-like receptor 3 (TLR3)). RLR signalling is inhibited by the cleavage and inactivation of MAVS by HCV NS3/4A[28,29]. In contrast, TLR3, which signals through the adaptor protein TRIF, senses HCV dsRNA, induces expression of interferon-stimulated genes (ISGs), which subsequently reduce viral replication[30–33]. Interestingly, the secretion of HCV dsRNA from the endosome modulates TLR3 activation, thereby moderating viral replication levels and potentially contributing to viral persistence[32]. However, the mechanism by which HCV dsRNA is delivered to endosome-localized TLR3 is incompletely understood.

Here, we used a TLR3-based directed evolution approach aiming at identifying viral determinants contributing to TLR3 delivery of HCV dsRNA. We identified a single point mutation in NS3h (D1467G) that attenuated replication and abrogated PRR activation by reducing dsRNA abundance. This function of D1467 was conserved amongst different genotypes. In addition, D1467G abrogated infectious particle production, potentially explaining the high conservation of this residue in vivo. Furthermore, we found that D1467G affected (−) strand synthesis and reduced interaction with NS5B polymerase, hinting towards a specific role of NS3h in (−) strand synthesis. This was confirmed by AlphaFold predictions, suggesting how NS3h and NS5B could be recruited, through known cis-acting RNA elements, to the 3'UTR, where a complex between NS3h, NS5B and the terminal stem-loop would be involved in priming (−) strand synthesis.

## Results

### A TLR3-based directed evolution approach identifies an NS3 helicase point mutant attenuating HCV replication and ablating TLR3 activation

Since TLR3 is the main sensor for dsRNA in HCV infected cells, leading to ISG induction and reduction of HCV replication[30–33], we performed a directed evolution approach aiming at identifying viral determinants of TLR3 activation (Fig. 1A). To this end, Huh7 cells persistently replicating a selectable subgenomic replicon (Huh7 neoJFH1[34]), under continuous selection with the antibiotic G418[35], were transduced with a lentiviral vector encoding TLR3. We hypothesized that a strong ISG response would lead to HCV eradication from the cell, which would consequently lead to a loss of G418 resistance and cell death. Therefore, HCV would only be able to replicate continuously through the mutation of viral or host determinants affecting TLR3 activation, thereby conferring an escape from TLR3 and allowing cell survival. Indeed, massive cell death was observed in Huh7 neoJFH1 cells expressing TLR3 but not in empty control cells. A few surviving cells could be expanded into clonal populations. Interestingly, a single point mutation in the helicase domain of NS3, D1467G, was detected in all sequences of TLR3 expressing independent cell clones, but not in empty control cells in two independent experiments, indicating a

strong selection advantage of the corresponding mutant in TLR3 expressing cells (Fig. 1B, Supplementary Fig. 1). In addition to D1467G, no other dominant mutations were detected.

To analyse the effect of D1467G on replication and TLR3 activation, we used luciferase reporter replicons[36], transfected into Huh7-Lunet empty or TLR3 expressing cells (Fig. 1C). TLR3 protein expression was confirmed by WB (Supplementary Fig. 2). D1467G had strongly delayed replication kinetics, compared to wt, but was not affected by the presence of TLR3 (Fig. 1D). In contrast, JFH1 wt replication reached ca. 10-fold higher levels than D1467G in empty cells at 96 h post-transfection but started to decline in TLR3 expressing cells starting at 48 h, reaching similar levels as D1467G at 72 and 96 h (Fig. 1D). JFH1 wt replication in TLR3 expressing cells was associated with a substantial induction of IFIT1 mRNA expression, indicative of a strong innate immune response counteracting replication, in line with previously published reports[30–33] (Fig. 1E). Strikingly, JFH1 D1467G did not induce IFIT1 expression, indicating that D1467G indeed abrogated TLR3 activation.

Since we could not exclude that the reduced TLR3 activation was in part due to the reduced replication of JFH1 D1467G, we aimed to analyse TLR3 activation in stable replicon cell lines. Therefore, selectable neoJFH1 wt/D1467G replicons were used to establish persistent cultures. To prevent reversion of D1467G to wt, we included two mutations at the nucleotide level still encoding a glycine (A4741G C4742G). We confirmed genomic stability by sequencing 6 weeks post-transfection (Supplementary Fig. 3). Stable neoJFH1 wt/D1467G were then transduced with lentiviral vectors, either empty or encoding TLR3 (Fig. 1F). HCV RNA levels were very similar in neoJFH1 wt compared to D1467G and differed only about 2-fold in empty vector transduced cells (Fig. 1G). Interestingly, JFH1 wt replication strongly declined upon TLR3 expression, while D1467G replication remained unaffected. Consistently, IFIT1 expression was induced around 15-fold in TLR3 expressing neoJFH1 wt cells (Fig. 1H), while D1467G did not induce detectable IFIT1 expression.

Overall, the results indicated that D1467G, a mutation in the helicase domain of NS3, abrogated TLR3 activation.

### The function of D1467 affects infectious particle production and is conserved across HCV genotypes

D1467G was a potential immune escape mutation, and we assumed that this mutation might be frequently found in clinical isolates. Therefore, we performed a sequence alignment containing 2555 full-length HCV sequences of all genotypes derived from the HCV GLUE database (Supplementary Fig. 4). Strikingly, D1467 was almost invariant, with 2553 sequences containing an aspartate and two chemically similar glutamates at this position (Fig. 2A).

HCV RNA contains several functional elements in the coding region[37]. To analyse whether the observed effect of D1467G was due to changes in amino acid or nucleotide sequence, we assessed the conservation on RNA level (Fig. 2B). The respective nucleotide A4741, which was mutated to G4741 in our screen, was also highly conserved, since an adenine at this position is strictly required to encode for aspartate. However, the third nucleotide of this and surrounding codons shows higher genetic flexibility, indicating that the amino acid rather than the nucleotide sequence is critical at this position of the genome.

The impact of D1467G on RNA replication appeared too moderate to explain the high degree of conservation; we thus hypothesized that this mutation might also affect other steps of the replication cycle, in particular since NS3 helicase is required for infectious particle production[16–24]. To study the effect of D1467G on virion assembly and release, we used a full-length chimeric virus encoding a Renilla luciferase reporter gene (JcR2A)[38] (Fig. 2C). Replication-deficient ΔGDD and assembly-deficient ΔE1E2 were included as controls. In vitro transcribed (ivt) RNAs corresponding to these variants were

 

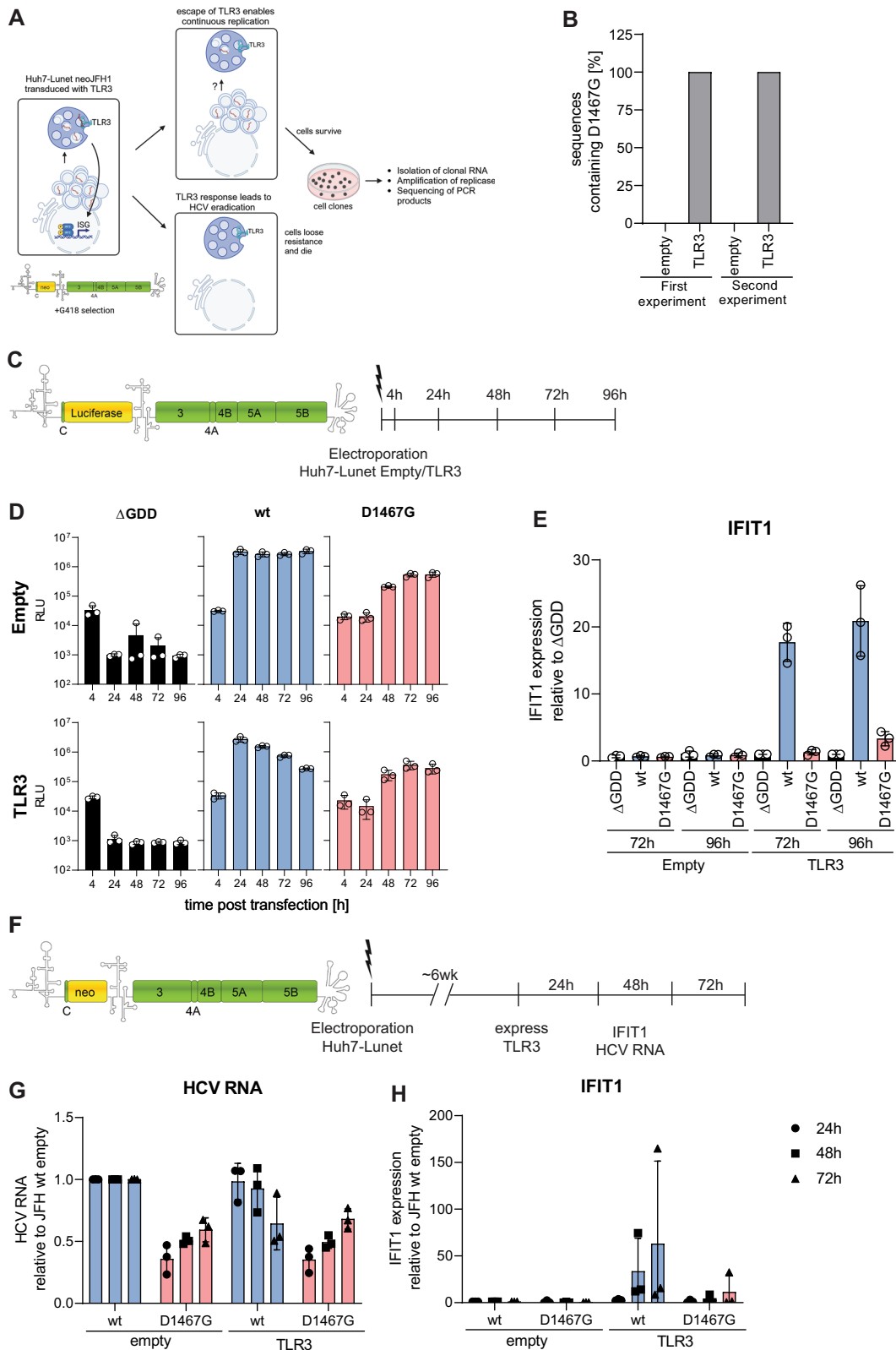

transfected into Huh7-Lunet CD81$^{high}$ cells, and luciferase activity in lysates was quantified as a correlate of RNA replication. Transfer of supernatants to naïve cells was used to determine infectious particle production. In line with previous experiments, D1467G reduced RNA replication about 7-fold compared to wt, yet being about 83-fold higher than ΔGDD at 72 h post-transfection (Fig. 2D). In contrast, luciferase activity in infected cells was approximately 65-fold lower for

mutant D1467G than wt infected cells and comparable to ΔGDD or ΔE1E2, suggesting that D1467 completely abrogated virus production, in addition to reducing RNA replication.

So far, all experiments were performed using the replicase of isolate JFH1 (gt2a). We therefore aimed to analyse the function of the equivalent residue D1463 in the most prevalent gt1 using the highly replicating GLT1E isolate (gt1b)[39]. In line with the results obtained for

**Fig. 1 | A TLR3-based directed evolution approach identifies an NS3 helicase point mutant attenuating HCV replication and ablating TLR3 activation.**
**A** Experimental scheme. Huh7 cells harbouring a persistent, selectable subgenomic replicon (neoJFH1) were transduced with lentiviral vectors encoding TLR3 or empty control and cultured until cell death, and single cell clones emerged. RNA was isolated from clonal cell populations, and the region encoding NS3-5B was amplified by RT-PCR. The PCR fragment was then directly sequenced using Sanger sequencing to identify conserved mutations. **A** was created in BioRender. Rothhaar, P. (2025) https://BioRender.com/j47e551. **B** Percentage of sequences containing D1467G in TLR3 or empty vector transduced cells in two independent experiments. **C**–**E** Huh7-Lunet empty/TLR3 cells were electroporated with in vitro transcribed (ivt) RNA of JFH1 wt/D1467G subgenomic reporter replicons, compared to a replication-deficient control (ΔGDD). **C** Schematic representation of the subgenomic reporter replicon and experimental setup. Coding sequences of HCV NS3-

NS5B and firefly luciferase are shown by green and yellow boxes, respectively; 5'UTR, EMCV-IRES, and 3'UTR are indicated by their RNA secondary structures. **D** Luciferase activity at the indicated time point was monitored as a correlate for RNA replication ($n = 3$ independent biological replicates). **E** IFIT1 mRNA expression relative to respective ΔGDD control was quantified by RT-qPCR ($n = 3$ independent biological replicates). **F**–**H** Selectable JFH1 subgenomic replicons encoding neomycin-phosphotransferase (neo) were used to generate Huh7 JFH1 wt/D1467G stable cell lines. Stable cell lines were cultured for approximately six weeks and transduced with TLR3-encoding lentiviral vectors or empty controls. **F** Schematic representation of a selectable bicistronic replicon and experimental setup (see legend to panel **C**). At indicated time points, RNA was harvested and HCV RNA (**G**) and IFIT1 levels (**H**) were quantified relative to the respective wt empty control by RT-qPCR ($n = 3$ independent biological replicates). **D**, **E**, **G**, and **H** Mean value is indicated by the bar graph and standard deviation is indicated by error bars.

JFH1, D1463G reduced GLT1E replication in Huh7-Lunet empty and TLR3 expressing cells (Fig. 2E) and abrogated TLR3 activation (Fig. 2F).

These data demonstrate that the function of D1463/D1467 is conserved amongst different HCV genotypes. Ablation of infectious virus production by D1467G furthermore hints towards a dual function of D1467 in replication and assembly/release, potentially explaining the high conservation of D1467 in vivo.

## D1467G affects TLR3 and RLR activation by reducing dsRNA levels

To assess whether D1467G specifically affected TLR3 activation, we next analysed the effect of D1467G on dsRNA sensing by RLRs, RIG-I and MDA5. HCV replication intermediates, in particular dsRNA, are generally sensed by RLRs[40–43], despite the recently discovered FAD-capping of the viral genomes[44]. However, activation of RLR signalling is inhibited by very efficient HCV NS3/4A-mediated MAVS cleavage[28,29], which was not affected by D1467G (Fig. 3A). To be able to analyse RLR activation by HCV, we therefore used Huh7 cells overexpressing MAVS containing the C508R mutation preventing NS3/4A-mediated cleavage (MAVScr)[45]. Similar to Huh7-Lunet TLR3 cells, JFH1 wt replication declined at 72 and 96 h in Huh7 MAVScr (Fig. 3B). Replication was attenuated by D1467G but constantly increased, reaching comparable levels to wt at 72 and 96 h. Strikingly, IFIT1 expression was induced by JFH1 wt about 150-fold, explaining the decreased replication at later time points, but not by JFH1 D1467G, suggesting that D1467G indeed also ablated RLR activation (Fig. 3C).

Taken together, these data suggested that D1467G abrogated activation of all PRRs sensing dsRNA.

Since the previous results indicated that D1467G abrogated activation of PRRs sensing dsRNA, we next analysed the effect of D1467G on dsRNA abundance. We used the persistent replicon cell lines neoJFH1 wt compared to D1467G due to their comparable steady state HCV RNA levels (Fig. 3D). HCV NS3 and dsRNA were detected by immunofluorescence (Fig. 3E) and quantified by automated image analysis (Fig. 3F). Comparable to HCV RNA levels, NS3 levels were reduced by about 50% in D1467G compared to wt but remained higher than the background signal in naïve cells. In contrast, dsRNA intensity was reduced in D1467G relative to wt replicating cells, with signal intensities approaching those in naïve cells, potentially indicating that dsRNA levels were specifically reduced. Since differences in background staining intensity and protein half-life hampered a direct comparison of dsRNA and NS3 levels, we aimed to validate these results by quantifying dsRNA in total RNA isolated from neoJFH1 wt and D1467G stable cell lines using a commercially available split-nanoluciferase-based assay. Interestingly, the fraction of dsRNA in total RNA isolated from D1467G cells was significantly lower compared to wt cells (Fig. 3G).

In summary, these data argued for a specific impact of the D1467G mutation on dsRNA abundance, thereby strongly reducing the activation of dsRNA-sensing RLRs and TLR3.

## D1467G reduces viral negative-strand synthesis

HCV (−) strand is far less abundant than (+) strand[46–48] and therefore the limiting factor of dsRNA formation. We therefore hypothesized that D1467G affected dsRNA abundance by reducing (−) strand levels. The highly conserved and structured 3'X tail is the presumptive promoter for the (−) strand synthesis, whereas the complementary sequence to the 5'UTR regulates (+) strand synthesis[49,50]. Previous work has shown that intergenotypic JFH1 replicon chimeras harbouring the 5'UTR (JFH/5'Con) or 3'X (JFH/3'XCon) from gt1b (isolate Con1) specifically affected (+) or (−) strand synthesis, respectively[46] (Fig. 4A). We therefore used these chimeras as controls for strand-specific effects of mutant D1467G in this study. First, replication and IFIT1 induction were analysed in transfected Huh7-Lunet cells in the presence and absence of TLR3 (Fig. 4B and C). JFH/5'Con and JFH/3'XCon replicated to similar levels and were slightly less attenuated than D1467G compared to their respective wt-controls (Fig. 4B). Expression of TLR3 mainly affected replication of the wt-controls, corresponding to their strong induction of IFIT1 expression already at 72 h post-transfection. Strikingly, IFIT1 induction was almost undetectable for JFH/3'XCon, known to be defective in (−) strand synthesis, thereby phenocopying mutant D1467G. In contrast, JFH/5'Con, despite replicating with the same efficiency as JFH/3'X Con, substantially increased IFIT1 induction up to JFH1 wt levels at 96 h (Fig. 4C). DsRNA levels correlated with IFIT1 induction, with JFH1 wt showing the highest and JFH/5'Con intermediate dsRNA levels. JFH/3'XCon and D1467G displayed the lowest level of dsRNA, comparable to the ΔGDD control (Fig. 4D).

Since these data suggested that the D1467G mutation affected dsRNA abundance by reducing (−) strand synthesis, we performed Northern blot analysis using strand-specific RNA probes (Fig. 4E). (−) strand RNA abundance was indeed remarkably reduced for mutant D1467G compared to the wt-controls, to the same extent as JFH/3'XCon. In line with lower (−) strand RNA levels, the (−)/(+) ratio was significantly reduced in D1467G and JFH1/3'XCon transfected cells compared to wt controls (Fig. 4F). Taken together these data indicated that mutation D1467G in the HCV helicase caused a drop in dsRNA abundance by specifically affecting (−) strand RNA synthesis.

The 3'X tail is supposed to promote initiation of negative-strand synthesis by the RdRp. The fact that a mutation in the helicase phenocopied the impact of a suboptimal, intergenotypic promoter in mutant JFH/3'XCon suggested a role of the helicase in the initiation of (−) strand synthesis. The relevant promoter mutations were located in the tip of the 3'terminal stem-loop SL1[46] and have been shown to affect negative strand synthesis by disturbing recognition of a genotype-specific sequence in the tip of SL1 by NS5B[46]. To support the claim that the NS3 helicase might be involved in the initiation of negative-strand synthesis, we searched for additional mutants phenocopying JFH/3'XCon and D1467G by a potentially different mechanism. The terminal bases of the HCV genome are part of the template that governs the synthesis of a priming di- or trinucleotide by NS5B, initiating with

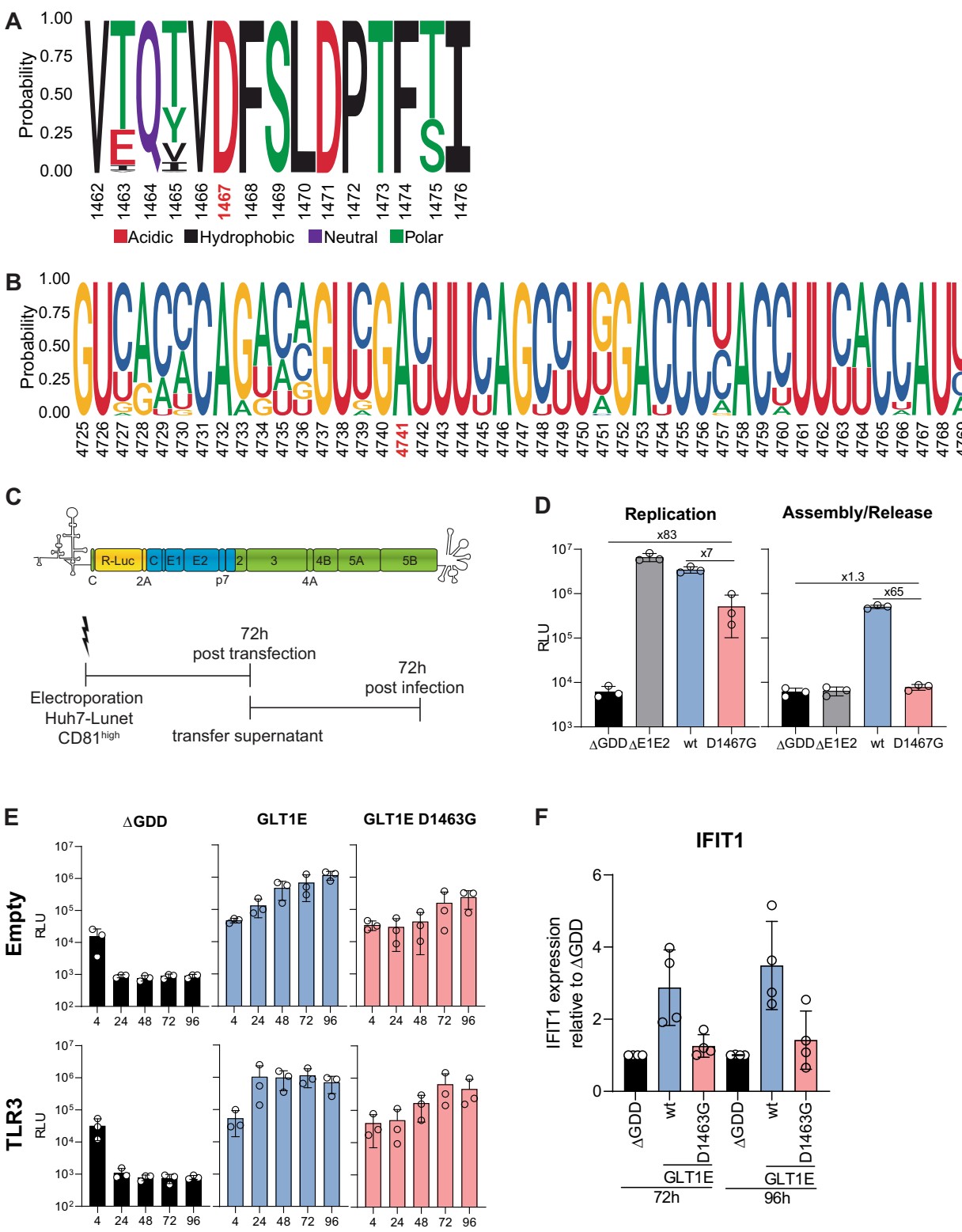

FAD[44], which is then used as a primer for elongation[51]. While the terminal U is invariant in cell culture due to the need of FAD-complimentarity, we mutated the penultimate G to U. This G2U mutation should disturb base pairing in the last stem-loop SL1 of the 3′X (Fig. 5A). Although the template requirements of primer synthesis are not fully understood so far, it seemed likely that G2U would also affect initiation of (−) strand synthesis by reduced efficiency of priming by NS5B. This is supported by a previous study, which reported that a G2C mutation

severely impaired replication, even when compensatory mutations restoring complementarity were introduced[52]. We therefore analysed the effect of this G2U mutation on replication, IFIT1 expression and negative strand RNA synthesis in comparison to wt and D1467G (Fig. 5A–E). Interestingly, G2U replicated to similar levels as D1467G, abrogated TLR3 activation comparable to D1467G (Fig. 5B and C) and reduced (−) strand abundance to comparable levels as D1467G (Fig. 5D and E).

**Fig. 2 | The function of D1467 affects infectious particle production and is conserved across HCV genotypes. A** and **B** Sequence logo of amino acids (**A**) or nucleotides (**B**) in proximity to D1467/A4741 generated by alignment of 2555 HCV-full length sequences from the HCV GLUE database. D1467 and A4741 are highlighted in red. **C** and **D** D1467G ablates infectious particle production. **C** Huh7-Lunet CD81[high] were electroporated with Jc1 full-length genomes encoding a Renilla luciferase reporter gene (JcR2A)[38]. Electroporated cells were harvested after 72 h to monitor replication. The supernatant obtained 72 h after transfection was transferred onto naïve Huh7-Lunet CD81[high] cells. Infected cells were harvested 72 h later to monitor particle production in electroporated cells (*n* = 3 independent

biological replicates). **D** JcR2A replication in transfected and infected cells was analysed by luciferase assay (*n* = 3 independent biological replicates). **E** and **F** Huh7-Lunet empty/TLR3 cells were electroporated with ivt RNA of HCV gt1b GLT1E wt/D1463G subgenomic reporter replicons, compared to a replication-deficient control (Con1 ΔGDD). **E** Luciferase activity at the indicated time point was monitored as a correlate for RNA replication (*n* = 3 independent biological replicates). **F** IFIT1 mRNA expression in Huh7-Lunet TLR3 fold respective ΔGDD control was quantified by RT-qPCR (*n* = 4 independent biological replicates). **D–F** Mean value is indicated by the bar graph, standard deviation is indicated by error bars.

---

In conclusion, our data demonstrate that a mutation in the NS3 helicase specifically reduced viral negative-strand synthesis, likely by affecting the initiation step.

## D1467G does not affect NS3 helicase or ATPase activity

We next analysed whether D1467G would affect NS3 helicase activity in vitro, thereby explaining the impact of this mutation on (−) strand synthesis. We focused on expression and purification of the NS3 helicase domain (NS3h, NS3 amino acids 188-632), which is sufficient for studying helicase activity in vitro[12,13,53–56]. To that end, JFH1 wt and D1467G NS3h proteins were expressed in BL21 (DE3) cells as C-terminal His-tagged proteins and purified by sequential Ni-affinity, gel-filtration, and anion exchange chromatography. Helicase unwinding activity was analysed using a fluorescence-based assay[56] (Fig. 6A). A shorter DNA oligo labelled with a 5′ Cy5 and 3′ Iowa Black RQ quencher was annealed with a longer unlabelled DNA oligo. After ATP-dependent unwinding by the NS3h, the short oligos form hairpins, bringing fluorophore and quencher in spatial proximity and quenching fluorescence (Fig. 6B). Unwinding rates were comparable for JFH1 wt and D1467G (Fig. 6C). In addition to separating duplex nucleic acids, NS3h hydrolyses ATP in a reaction stimulated by the presence of nucleic acids (Fig. 6D). Both proteins catalysed ATP hydrolysis at similar rates, both in the presence and absence of RNA (Fig. 6E). Unlike other helicases, HCV NS3h displays conserved nucleic acid sequence specificity[54], which appeared to be unaffected by the D1467G substitution (Fig. 6F). For mutant and wildtype NS3h proteins, pyrimidine sequences stimulated more efficiently than purine sequences. Notably, both proteins interacted similarly with RNA containing a triphosphate at the 5′end, the PAMP detected by IFIT proteins[57], and the 3′end of the viral genome SL1 (Fig. 6F). We further assessed binding of NS3h to different SL1-derived RNA and DNA oligonucleotides in a fluorescence polarization-based in vitro assay[58]. In agreement with literature[59], binding affinities were highest with ssDNA, followed by dsDNA and partial ssRNA and almost no binding to dsRNA, represented by SL1, with no difference between wt and D1467G for any variant RNA and DNA oligonucleotides (Supplementary Fig. 5). Taken together, these data indicate that D1467G did not affect NS3h substrate binding or helicase or ATPase activity in vitro.

## D1467G disturbs the interaction between NS3 and NS5B

Due to the lack of impact of D1467G on the biochemical activity of the helicase in vitro, we next aimed to find alternative mechanisms explaining the defect caused by this mutation in (−) strand synthesis.

Upon analysis of wt and mutant NS3 in replicon cells by SDS-PAGE and Western blot, we found that the D1476G mutant NS3 had an apparently lower molecular weight than the wt-protein (Fig. 7A and B), pointing towards changes in post-translational modifications, which might regulate the function of NS3h. However, NS3 from replicon cells, ectopically expressed NS3 in the context of NS3-5B and NS3/4A, all showed the same difference. Consistently, the increased mobility in SDS-PAGE for the D1467G mutant was also observed with bacterial-expressed NS3/4A (Fig. 7B), rendering changes in post-translational modifications unlikely, but suggesting an aberrant running behaviour caused by the mutation.

In addition to the increased mobility of NS3, D1467G caused a change in the NS5A phosphorylation pattern in replicon cells and upon expression of NS3-NS5B (Fig. 7B and C). NS5A exists in two phospho-isoforms, termed p56/p58 or hypo- and hyperphosphorylated, that can be visualized as two discrete bands in SDS−PAGE/WB. While it is clear that the phosphorylation status is critical for viral replication and assembly, it has so far not been possible to assign specific functions to p56 or p58. However, determinants affecting p58 synthesis have been found in all viral NS-proteins and seem to point to changes in interactions within the replicase[60]. A significant reduction in p58 abundance associated with D1467G, resulting in a change in p56/p58 ratios, was not only found in replicon cells, but also upon ectopic expression of NS3−NS5B (Fig. 7A−C). Another mutation in NS3h, A1655G, was recently published to reduce NS5A hyperphosphorylation without affecting replication[24]. Indeed, A1655G reduced NS5A hyperphosphorylation to a similar extent as D1467G (Fig. 7D, Supplementary Fig. 6A). However, A1655G did neither affect replication, as reported, nor TLR3 activation (Fig. 7E, Supplementary Fig. 6B), suggesting that the change in NS5A hyperphosphorylation might not be the underlying cause for defective (−) strand synthesis of mutant D1467G.

Since the reduction of NS5A hyperphosphorylation indicated changes in replication complex formation, we performed co-immunoprecipitation (Co-IP) to analyse the effect of D1467G on the interaction of NS3 with other NS proteins. NS3-5B non-structural proteins were expressed transiently in Huh7-Lunet T7 cells, NS3-IP was performed, and NS proteins were detected by WB in the presence and absence of the cell-permeable crosslinker DSP (Fig. 7F and G, Supplementary Fig. 7). We found detectable interaction of NS3 with NS4B, NS5A, and NS5B in the presence of DSP, which was not affected by the D1467G mutation (Supplementary Fig. 7). In the absence of DSP, we failed to detect a co-precipitation of NS4B and NS5A, but, importantly, found that D1467G significantly reduced Co-IP of NS5B (Fig. 7F and G). These data suggest that D1467G affected the strength or the duration of the interaction between NS3 and NS5B, likely contributing to the defect of the mutant in viral (−) strand RNA synthesis.

## eCLIP reveals NS3 binding to defined RNA elements in viral (+) and (−) RNA

In addition to affecting NS3 and NS5B interaction, we next aimed to analyse whether D1467G impacted the binding of NS3 to HCV RNA. We mapped NS3 RNA interaction using enhanced crosslinking and immunoprecipitation in combination with RNA sequencing (eCLIP-seq)[61]. We normalized the signal observed in NS3 immunoprecipitations with respect to sequencing depth and performed an enrichment analysis comparing the normalized coverage relative to a size-matched input (SMI) control as previously described[62]. We observed several significantly enriched peak regions (adjusted *P* value < 0.05, log$_2$-fold change > 2; Supplementary Data 1) that were largely restricted to the 5′ and 3′ end regions of (+) RNA (Fig. 8A). Surprisingly, when analysing NS3 binding to (−) RNA, we observed only a single significantly enriched peak (adjusted *P* value < 0.05, log$_2$-fold change > 2; Supplementary Data 1) located at the immediate 5′ end of the (−) RNA strand precisely overlapping the previously described SLα (Fig. 8A−C)[63]. Strikingly, NS3 binding sites in (+) RNA overlapped two stem loops in the NS5B coding

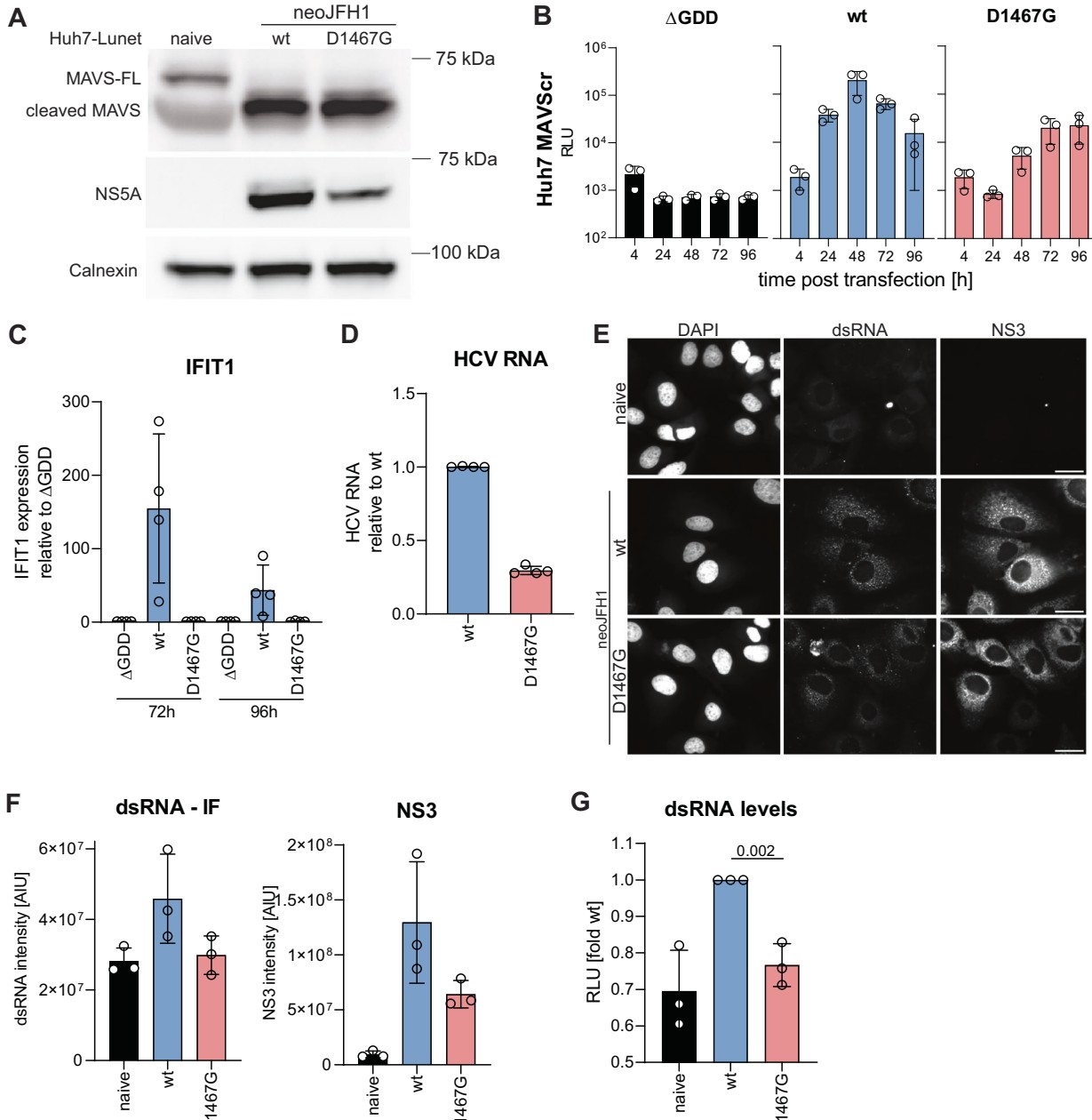

**Fig. 3 | D1467G ablates RLR signalling and reduces the abundance of dsRNA.**
**A** Whole cell lysates of Huh7-Lunet naïve and neoJFH1 wt/D1467G were analysed regarding NS5A, MAVS and Calnexin expression by WB. Representative of three independent experiments. B-C) Huh7 MAVS KO MAVScr cells, expressing the cleavage-resistant MAVS-mutant C508R, were electroporated with ivt RNA of JFH1 wt/D1467G subgenomic reporter replicons, compared to a replication-deficient control (ΔGDD). **B** Luciferase activity at the indicated time point was monitored as a correlate for RNA replication ($n = 3$ independent biological replicates). **C** IFIT1 mRNA expression fold respective ΔGDD control was quantified by RT-qPCR ($n = 4$ independent biological replicates). **D** HCV RNA levels in Huh7-Lunet neoJFH1 wt/D1467G relative to wt cells were quantified by RT-qPCR ($n = 4$ independent

biological replicates). **E** Huh7-Lunet naïve and neoJFH1 wt/D1467G were stained for dsRNA and NS3. Representative of $n = 3$ independent experiments. Scale bar indicates 30 μm. **F** At least 364 imaged cells per condition and independent replicate ($n = 3$ independent biological replicates) were segmented using cellpose, and dsRNA and NS3 per cell intensity were quantified using FIJI. Mean of all cells of each replicate is graphed. **G** dsRNA levels in total RNA isolated from neoJFH1 wt/D1467G cells ($n = 3$ independent biological replicates) were analysed using a commercially available dsRNA bioassay. Statistical analysis was performed by an unpaired two-sided $t$-test using GraphPad prism. $P$-values are indicated. **B**–**D**, **F**, and **G** Mean value is indicated by the bar graph and standard deviation is indicated by error bars.

region, 5BSL3.1 and 5BSL3.3 (Fig. 8B+C)[64,65], which are part of a large cruciform RNA structure. The proximal 5BSL3.2 was not bound by NS3 (Fig. 8B and C) but has been reported to be bound by NS5B[66] and to interact with the 3'X SL2 through kissing loop interaction[64,65], thereby positioning the 5BSL3 RNA structure in proximity to the 3' end of the genome. The apparent lack of significance of D1467G binding to 5BSL3.1 (Fig. 8B) was only due to a minimal difference in log₂FC values (wt: 2.5,

D1467G: 1.99, Supplementary Data 1), thereby failing our inclusion criteria (log₂FC > 2). We furthermore observed binding of NS3 to a multitude of cellular RNAs (Supplementary Data 2), including tRNAs, mRNAs, and ncRNAs. The target RNAs of WT and D1467G were overall widely overlapping. In most cases, fewer reads were found for the mutant, in agreement with its attenuated replication efficiency, resulting in reduced NS3 abundance (Fig. 3F).

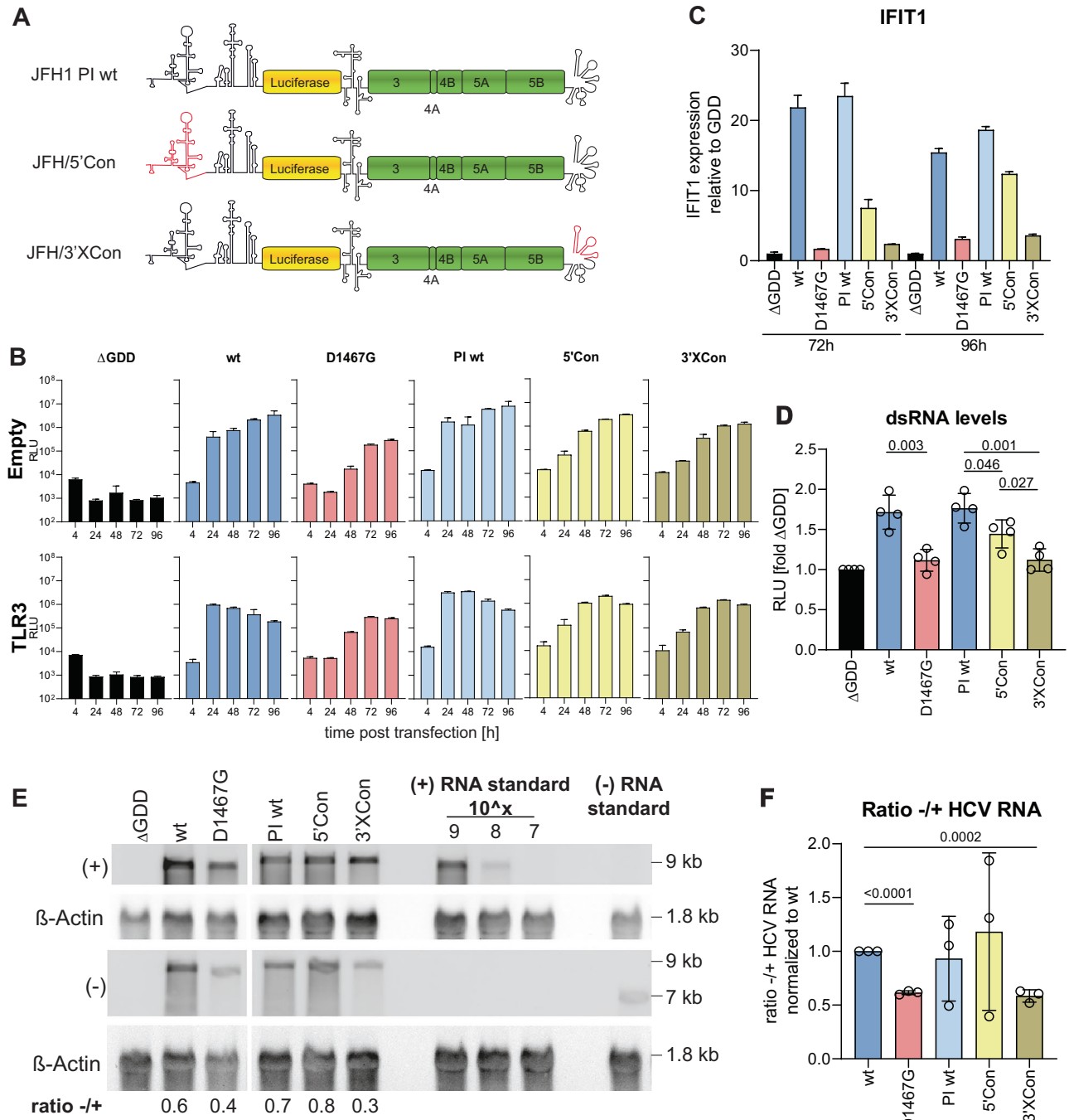

**Fig. 4 | D1467G specifically reduces (−) strand synthesis. A** Scheme of chimeric replicons based on JFH1 harbouring gt1b (Con1) 5' or 3'X sequences[46]. Sequences derived from Con1 are highlighted in red. Since luciferase translation is mediated by an additional poliovirus IRES (PI), (compare to scheme in Fig. 1C), the respective wt-construct was included as a reference (PI wt). **B**–**F** Huh7-Lunet empty/TLR3 cells were electroporated with indicated constructs, and replication at the indicated time point was monitored using luciferase reporter activity (RLU) (**B**) (*n* = 3 independent biological replicates). **C** IFIT1 mRNA expression was quantified by qPCR (*n* = 3 independent biological replicates). **D** dsRNA levels isolated from Huh7-Lunet empty cells 72 h post-transfection were analysed using a commercially available dsRNA bioassay (*n* = 4 independent biological replicates). **E** Effect of D1467G on (+)

and (−) strand levels was analysed by Northern blot. RNA was isolated from Huh7-Lunet empty cells 72 h post-transfection, and NB was performed using DIG-labelled strand-specific RNA probes. Band intensity of (−) and (+) HCV RNA was quantified using FIJI. Size of (+) and (−) HCV RNA, HCV standards and ß-Actin is indicated. **F** Ratio (−)/(+) RNA in NB was quantified (*n* = 3 independent biological replicates) and normalized to the wt control. Note that all samples were analysed on the same blot, but some lanes not included in further analysis were removed for clarity. Statistical analysis was performed using an unpaired two-sided *t*-test. P-values are indicated. **B**–**D**, **F** Mean value is indicated by the bar graph, standard deviation is indicated by the error bars.

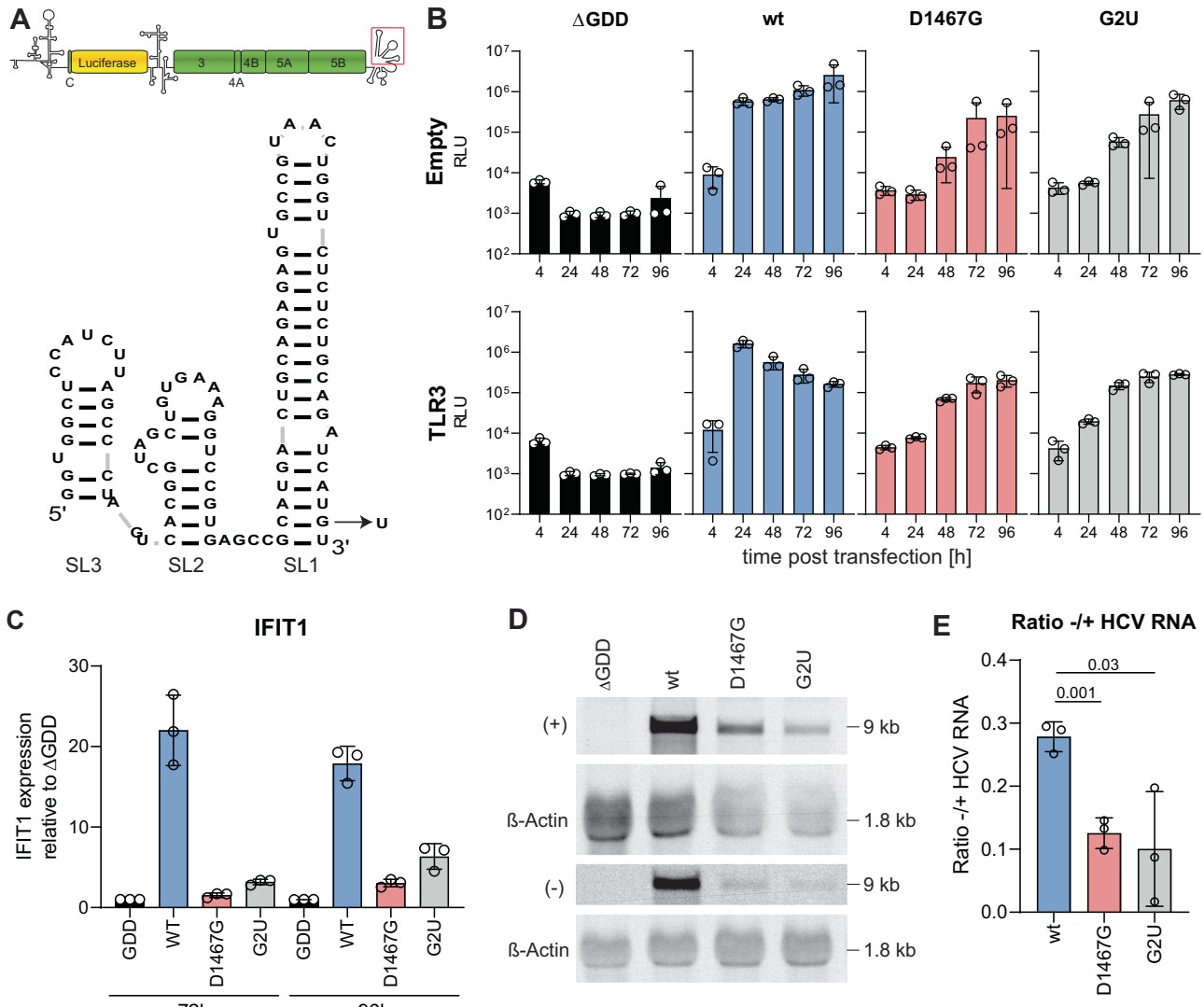

**Fig. 5 | Mutation of the 3'X G2U mimics D1467G. A** RNA structure of the X-tail visualized using RNA-based template visualization[125]. G2U transversion is indicated. **B** and **C** Huh7-Lunet empty/TLR3 cells were electroporated with ivt RNA of the indicated JFH1 constructs. **B** Luciferase activity at the indicated time point was monitored as a correlate for RNA replication (n = 3 independent biological replicates). **C** IFIT1 mRNA expression relative to the respective ΔGDD control was quantified by RT-qPCR. **D** (+) and (−) strand levels were analysed by Northern blot. RNA was isolated from Huh7-Lunet empty cells 72 h post-transfection. Band intensity of (−) and (+) HCV RNA was quantified using FIJI. Size of (+) and (−) HCV RNA and ß-Actin is indicated. **E** Ratio (−)/(+) RNA in NB was quantified (n = 3 independent biological replicates). Statistical analysis was performed using an unpaired two-sided t-test. P-values are indicated. **B**, **C**, and **E** Mean value is indicated by bar graph and standard deviation is indicated by error bars.

Overall, the eCLIP analysis identified NS3 binding sites near the 3'end of the genomic RNA and the 5'end of the negative strand, potentially contributing to the regulation of (−) strand RNA synthesis. However, in agreement with the lack of differential binding to DNA and RNA in vitro, we found no substantial differences between WT and D1467G that could contribute to the defect of the mutant in negative-strand synthesis.

## AlphaFold predictions of NS3h, RNA and NS5B suggest possible roles for D1467 in initiation of (−) strand synthesis

In view of NS3 binding to (+) 5BSL3 and (−) SLα (the counterpart of 3'X SL1) and of the reduced NS3 and NS5B interaction, we next aimed to model atomic structures of complexes of NS3 with NS5B and, by taking advantage of the newly available Alphafold3 (AF3)[67], with viral RNA. We tried various combinations of NS3 or NS3h and single-stranded or highly structured RNAs from 5BSL3 or from the 3'X, with or without

NS5B and with or without magnesium ions. We got two definite hits, as judged by AF3 predicted aligned errors (PAE) (Fig. 9A and E).

The first hit is a complex of NS3h with 5BSL3 (Fig. 9A–D). The single-stranded stretch of RNA between 5BSL3.2 and 5BSL3.3 (Fig. 9B and C) threads through the NS3h RNA binding groove, as previously described[68]. Remarkably, in this position, NS3h makes high-confidence contacts with 5BSL3.3. and 5BSL3.1, in agreement with the eCLIP data, while 5BSL3.2 is displayed on the side of the complex (Fig. 9A–C). Furthermore, the Phe-loop contributes to a negatively charged patch at the interface with D3 (Fig. 9C) through D1467 and D1471 (Fig. 9D), with D1467G reducing the electronegativity of this patch. However, while the AF3 prediction of NS3 binding to this region is compelling and supports the eCLIP data, previous studies showed no or only limited impact of mutations in SL3.1 and SL3.3, respectively, on RNA replication[69], suggesting rather auxiliary than essential functions of this interaction.

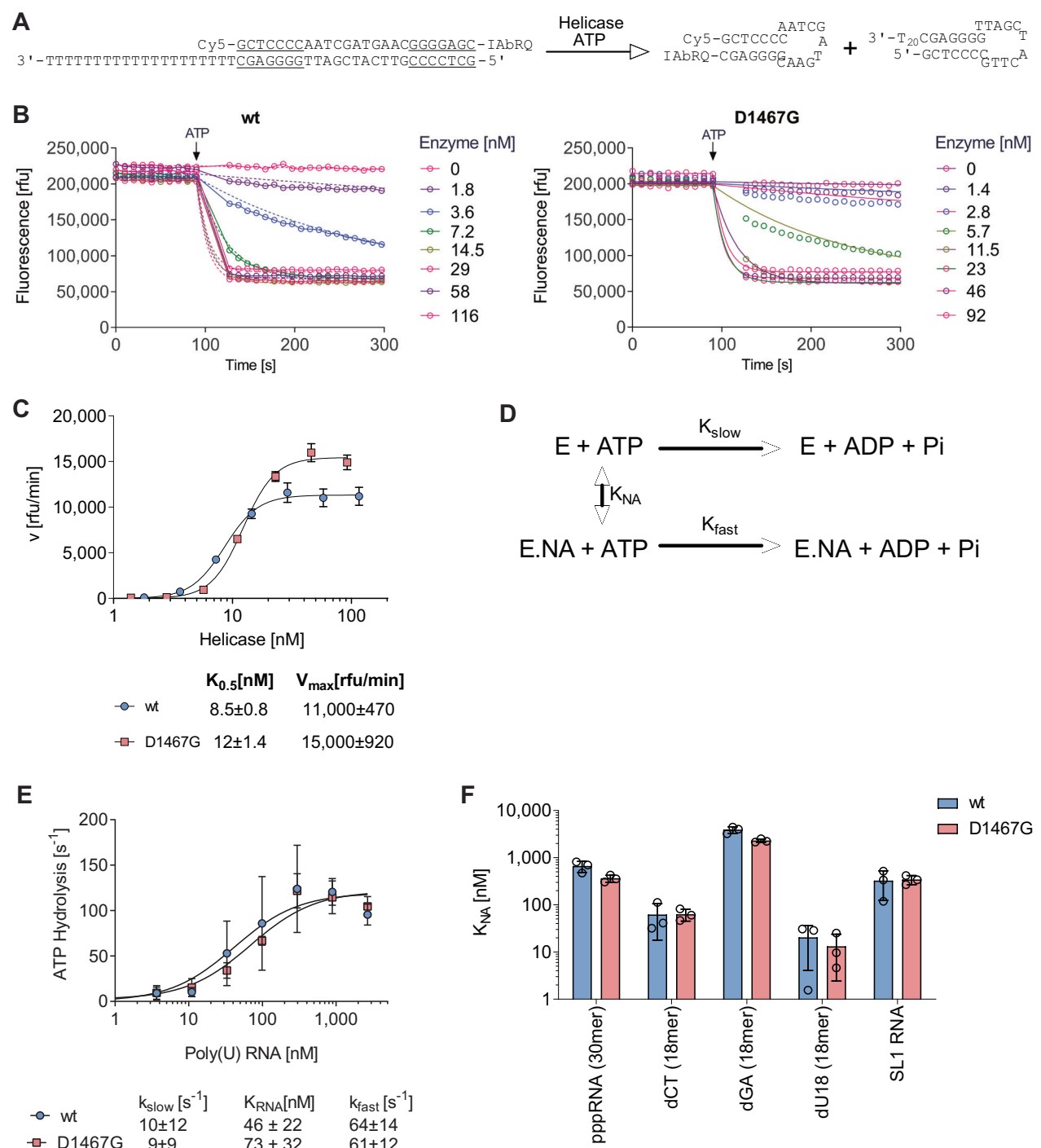

**Fig. 6 | D1467G does not affect NS3 helicase or ATPase activity. A** Scheme for the helicase assay where ATP-fuelled helicase-catalysed DNA duplex separation leads to the formation of a quenched molecular beacon[126]. **B** Ability of purified wt and D1467G NS3h to unwind DNA in a concentration-dependent manner. **C** Comparison of initial DNA unwinding rates observed in helicase assays in the presence of various concentrations of wt (circles) or D1467G (squares) NS3h. Average rates from 3 separate reactions are shown. Error bars are standard deviations. Data are fit to a concentration–response equation with GraphPad Prism. $K_{0.5}$ is the helicase concentration needed to stimulate unwinding to 50% of the maximum velocity ($V_{max}$). **D** Scheme for helicase-catalysed ATP hydrolysis. **E** Initial rates of NS3h-catalysed ATP hydrolysis in the presence of various concentrations of PolyU RNA. $n = 3$ independent biological replicates. **F** Concentrations of various oligonucleotides needed to stimulate ATP hydrolysis by 50%. $n = 3$ independent biological replicates. Averages and standard deviations (error bars) are shown.

The second hit was obtained with a combination of NS3h, NS5B, and the last 40–43 bases of the 3′X. Of note, 3′X can adopt different conformations, supposed to switch from translation to replication mode[70,71], regulated by the kissing loop interaction of SL2 with stem loop 3.2 in the NS5B coding region[64,72]. SL1, the terminal stem-loop

encompassing the last 46 bases of the genome, thereby undergoes minor changes to form SL1′, resulting in the 3′end either being a blunt end or containing a 3nt overhang, respectively[71,73]. This minor change might be critical to enable genome replication, with both NS3h and NS5B RdRP requiring short overhangs to initiate unwinding or RNA

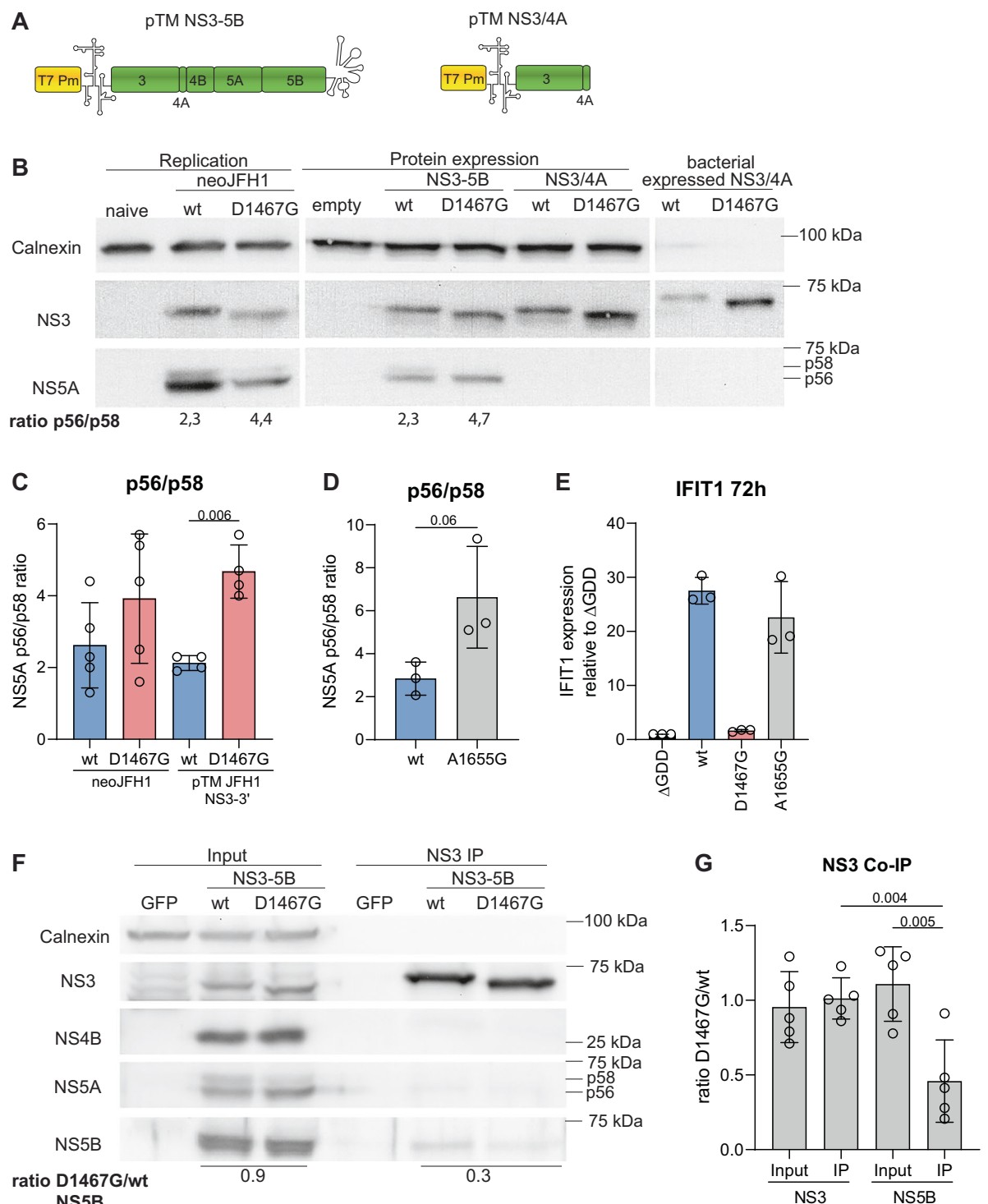

**Fig. 7 | D1467G reduces NS5A hyperphosphorylation and interaction of NS3 and NS5B. A** Schematic representation of the JFH1 constructs used for transient protein expression. Coding sequences of HCV NS3-NS5B or NS3/4A are indicated in green. The T7 promoter is indicated in yellow. EMCV-IRES and authentic 3'UTR are indicated by their RNA secondary structures. **B** NS3 and NS5A from replicon cells and upon ectopic expression in human cells were analysed by WB and compared to bacterially expressed NS3/4A. Huh7-Lunet T7 were used to express JFH1 NS3-5B or NS3/4A replication-independently. The band intensity of the p56 and p58 NS5A bands was quantified using FIJI. **C** Ratio of p56/p58 in Huh7-Lunet neoJFH1 wt/D1467G and Huh7-Lunet T7 cells transfected with the indicated construct (n = 5 independent biological replicates for neoJFH1, n = 4 independent biological replicates for pTM NS3-5B). **D** and **E** Huh7-Lunet empty/TLR3 cells were electroporated with JFH1 wt/D1467G/A1655G subgenomic replicon. **D** Effect on NS5A hyperphosphorylation in Huh7-Lunet empty was analysed by WB, and the p56/p58 ratio was quantified using FIJI (n = 3 independent biological replicates). **E** IFIT1 mRNA expression in Huh7-Lunet TLR3 was quantified relative to the respective ΔGDD control by RT-qPCR. **F** and **G** JFH1 NS3-5B wt/D1467G was transiently expressed in Huh7-Lunet T7 cells. **F** NS3 was immunoprecipitated, and the Co-IP of other HCV NS proteins was analysed by WB. Band intensity was measured using FIJI. Ratio of band intensity of NS5B D1467G/NS5B wt is indicated. **G** Ratio of respective D1467G to wt band (n = 5 independent biological replicates). **C**–**E**, **G** Statistical analysis was performed using an unpaired two-sided *t*-test using GraphPad Prism. *P*-values are indicated. Mean value is indicated by a bar graph and standard deviation is indicated by error bars.

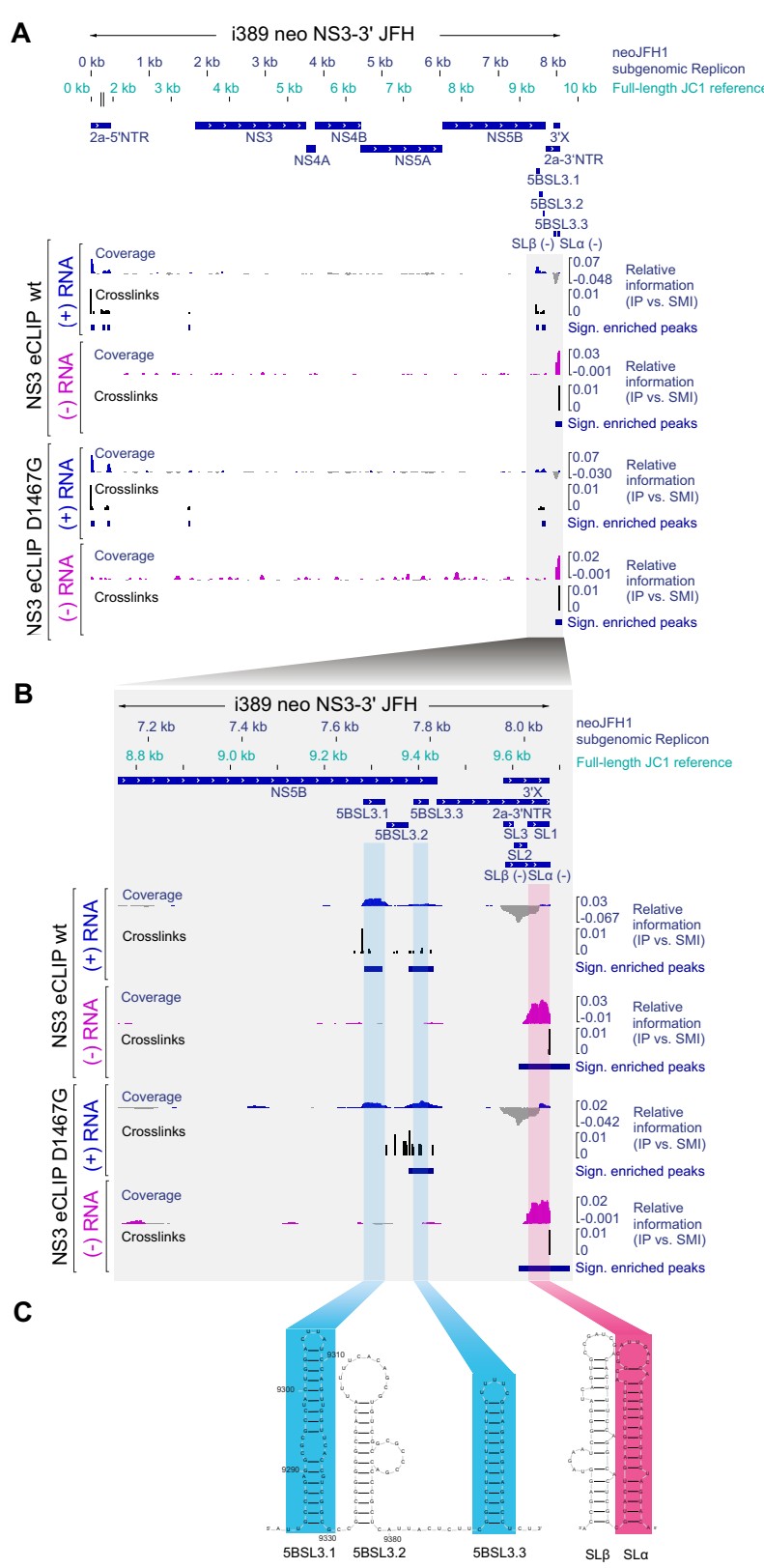

synthesis, respectively[74–76]. The best AF3 predictions comprising NS3h, NS5B, and SL1 were obtained with such 3' overhangs mimicking the SL1' conformation. In the best prediction as judged by PAE (Fig. 9E–G), the 3'overhang again threads through the NS3h RNA binding groove, its 5' going over the beta-hairpin in NS3h domain 2 as described previously[68]. The model shows that this beta hairpin, with D1467 near its tip, is thereby positioned to separate the two strands of the stem

upstream (Fig. 9F). The stem itself binds a pocket in the thumb of NS5B, an interaction reminiscent of the one reported between NS5 and stem-loop A of dengue virus[77] (Supplementary Fig. 8A–C). In the NS3h–NS5B interaction, charge complementarity appears important, with extensive contacts between a positive zone in the fingertips of NS5B and an extended negatively charged region of NS3h, including the negative patch and the C-terminus of D3. Thus, although there is no

**Fig. 8 | eCLIP reveals NS3 binding to defined RNA elements in viral (+) and (−) RNA. A** Alignment of NS3 eCLIP data for neoJFH1 wt/D1467G replicons to the corresponding neoJFH1 subgenomic replicon reference sequence. Location in subgenomic and full-length RNA is indicated. Relative coverage information in IP vs. SMI is calculated at each position and displayed for positive-strand RNA (blue) and negative-strand RNA (pink). One-sided Fisher's exact test was performed and Benjamini–Yekutieli[121] procedure was applied to correct for multiple testing. NS3 binding sites significantly enriched relative to SMI (adjusted $P$ value < 0.05, log₂-

fold change > 2) are displayed below the tracks. For each significantly enriched NS3 binding site, crosslinking events are extracted and shown in black (relative information in IP vs. SMI). **B** Zoomed-in view of NS3 eCLIP signal at the 3' end of the viral genome. Blue and pink boxes indicate significantly enriched NS3 binding sites overlapping known RNA secondary structures in (+) strand RNA (blue) and (−) strand RNA (pink). **C** Illustration of known RNA secondary structures at regulatory regions overlapping indicated NS3 binding sites.

direct contact between NS5B and D1467, the D1467G mutation may alter the NS3h–NS5B interaction by weakening long-range electrostatic interactions. Altogether, this second hit, comprising NS3h–SL1'–NS5B, may represent a 'priming complex', in which interactions between the three partners would position the 3'end of the (+) strand to be presented to the RNA binding groove of NS5B (Fig. 9F).

## Discussion

In this study, we identified a single NS3 point mutation, D1467G, which reduces (−) strand synthesis by disrupting the interaction of NS3 with NS5B. Our data, supported by AF3 predictions, suggest that NS3h, NS5B, and the terminal stem-loop of the HCV genome, SL1', form the initiation complex for negative-strand synthesis. NS3h, thereby, is supposed to unwind SL1' to render it accessible for de novo initiation of RNA synthesis by the RdRp NS5B. Our model is further supported by three independent mutants, resulting in very similar effects on viral negative strand synthesis: D1467G in NS3h affecting the interaction with NS5B, an intergenotypic X-tail chimera (3'X) disturbing sequence-specific interactions with NS5B and a mutation of the penultimate nucleotide of the HCV genome (G2U), likely affecting primer synthesis by NS5B. While this is no formal proof, it strongly suggests that NS3h plays an active role in the initiation of (−) strand synthesis. Since the NS3 helicase possesses 3'–5'-helicase activity, the 3' end of the X-tail is a plausible binding and initiation site of unwinding. This view is supported by our eCLIP data, demonstrating binding to 5BSL3 of genomic RNA, as well as binding to the 5'SLα of (−) strand RNA, all supposed to be critically involved in the initiation of negative-strand synthesis.

D1467 is located in the so-called Phenylalanine loop (Phe-loop), a ß-loop structure connecting the ß-hairpin stretching from D2 into D3 of HCV NS3. Experimental evidence determining the function of the Phe-loop is so far missing. D1467G decreases (−) strand synthesis and reduces helicase-polymerase interaction, shedding light on an unexplored role of the Phe-loop. In addition, D1467G abrogates infectious particle production, highlighting the dual function of this residue. Whether these phenotypes are linked to the general function of the Phe-loop is unclear. Reported Phe-loop mutations had very different effects on NS3h activity in vitro, ranging from abrogated helicase activity while maintaining ATPase activity (F438A), abrogated helicase and ATPase activity (F444A, both[78]) or reduced helicase activity (F438L[79]). Interestingly, ß-hairpin structures connecting D2 and D3 also occur in other helicases, e.g., in closely related BVDV or Ortho-flavivirus NS3. Structurally related stem loops can also be found in distantly related helicases like yeast Prp34p family, which members contribute to splicing and ribosome biogenesis[80], DHX36, which binds and unwinds specifically G-quadruplex structures[81], and archaeal DNA helicase Hel308. It has been hypothesized that the beta-hairpin structure of Hel308 could act as a plough to separate the two strands in RNA stems[82]. Our AlphaFold prediction of a putative complex between NS3h, SL1', and NS5B supports this hypothesis.

Due to the presence of conserved helicase motifs in viral genomes[83] it has been proposed that positive-sense RNA viruses with a genome larger than 7 kb encode a helicase[84], linking genome size to the necessity of a helicase for efficient viral replication. Helicase-motif containing proteins are found, e.g., in *Picornaviridae* (2C), *Coronaviridae* (nsp13), *Togaviridae* (nsp2), *Flaviviridae* (NS3), *Hepeviridae* (ORF1)

and others[83]. However, the presence of conserved helicase motifs does not necessarily implicate unwinding activity that can be detected in vitro[85], which has been demonstrated for *Coronaviridae* Nsp13 (5'–3')[86], Alphaviruses Nsp2 (5'–3')[87], *Flaviviridae* NS3 (3'–5')[88,89], hepatitis E virus ORF1 (5'–3')[90] and others. It is often proposed that helicases unwind complex RNA secondary structure or the dsRNA intermediate and therefore enable RdRP-mediated transcription, but experimental evidence supporting this is so far lacking for most, if not all, viral helicases. Due to the variety of functions of cellular helicase-like proteins, as well as the diversity of viral helicase-like proteins it appears reasonable to assume that the function of viral helicases is likely not conserved amongst unrelated viruses, but rather depends on the structure and directionality of the respective helicase-like protein, associated and regulatory domains (like protease domain for *Flaviviridae* NS3[55] or zinc binding domain for *Coronaviridae*[91] and *Picornaviridae*[92]) and viral genome organization. Structural data indicate that nsp13, the 5'–3' helicase encoded by *Coronaviridae*, is part of the replicase and might be important for backtracking in the proof-reading process during RNA replication[93–95]. In addition, Alphavirus Nsp2 is a 5'–3' helicase that is critical for the synthesis of subgenomic RNA[96]. However, the mechanism by which Nsp2 contributes to subgenomic RNA synthesis is still unclear. For pestiviruses, indeed, a role of the helicase in viral (−) strand synthesis has been proposed[97], but in the absence of a suggested mechanism.

Our eCLIP data indicate binding of NS3 to 5BSL3.1 and 5BSL3.3, while our AF3 prediction of the interaction suggests such a complex could help display 5BSL3.2, a hub of interactions regulating HCV (+) strand functions. Indeed, 5BSL3.2 has been reported to directly bind NS5B[66] and to regulate translation or replication of viral RNA through long-range interactions[98–100], interact with 3'X SL2[64,65], thereby positioning both CREs in spatial proximity. Therefore, the 5BSL3 structure may serve as an assembly site of an NS3-NS5B complex. Transfer of the NS3-NS5B complex to the 3'X enables NS3h-mediated unwinding of the 3'X and (−) RNA synthesis by NS5B[101], as indicated by AF3 predictions. Unwinding of the 3'X tail by HCV NS3h is a tempting model, and our data provide a hint towards distinct functions of viral helicases in the initiation of negative-strand synthesis.

## Methods

### Biological resources

All mammalian cell lines were cultured in Dulbecco's modified Eagle's medium (DMEM) supplemented with 10% foetal calf serum (Seromed, inactivated for 30 min at 56 °C), 100 U/mL penicillin and 100 µg/mL Streptomycin, 2 mM ʟ-glutamine (Gibco) and 1% 100x non-essential amino acids (Gibco) (DMEM complete) at 37 °C and 5% $CO_2$. Selected drugs were added to the media. Huh7 neoJFH1 (G418 1 mg/mL)[34], Huh7-Lunet[64], Huh7-Lunet empty/TLR3 (Blasticidin, 10 µg/mL)[32], Huh7 MAVS KO MAVScr (Blasticidin, 10 µg/mL)[45], Huh7-Lunet CD81[high] (G418 1 mg/mL)[102], and Huh7-Lunet T7[103] were published previously. Huh7-Lunet neoJFH1 wt and Huh7-Lunet neoJFH1 D1467G were generated in this study by electroporation of Huh7-Lunet with pFKi389neoJFH1_NS3-3' wt and D1467G in vitro transcribed (ivt) RNA and selection with 1 mg/mL G418 for six weeks.

*E. coli* DH5α[104] was used for plasmid amplification. *E. coli* BL21 (DE3) (Thermo Fisher Scientific, catalogue number EC0114) was used for IPTG-inducible protein expression.

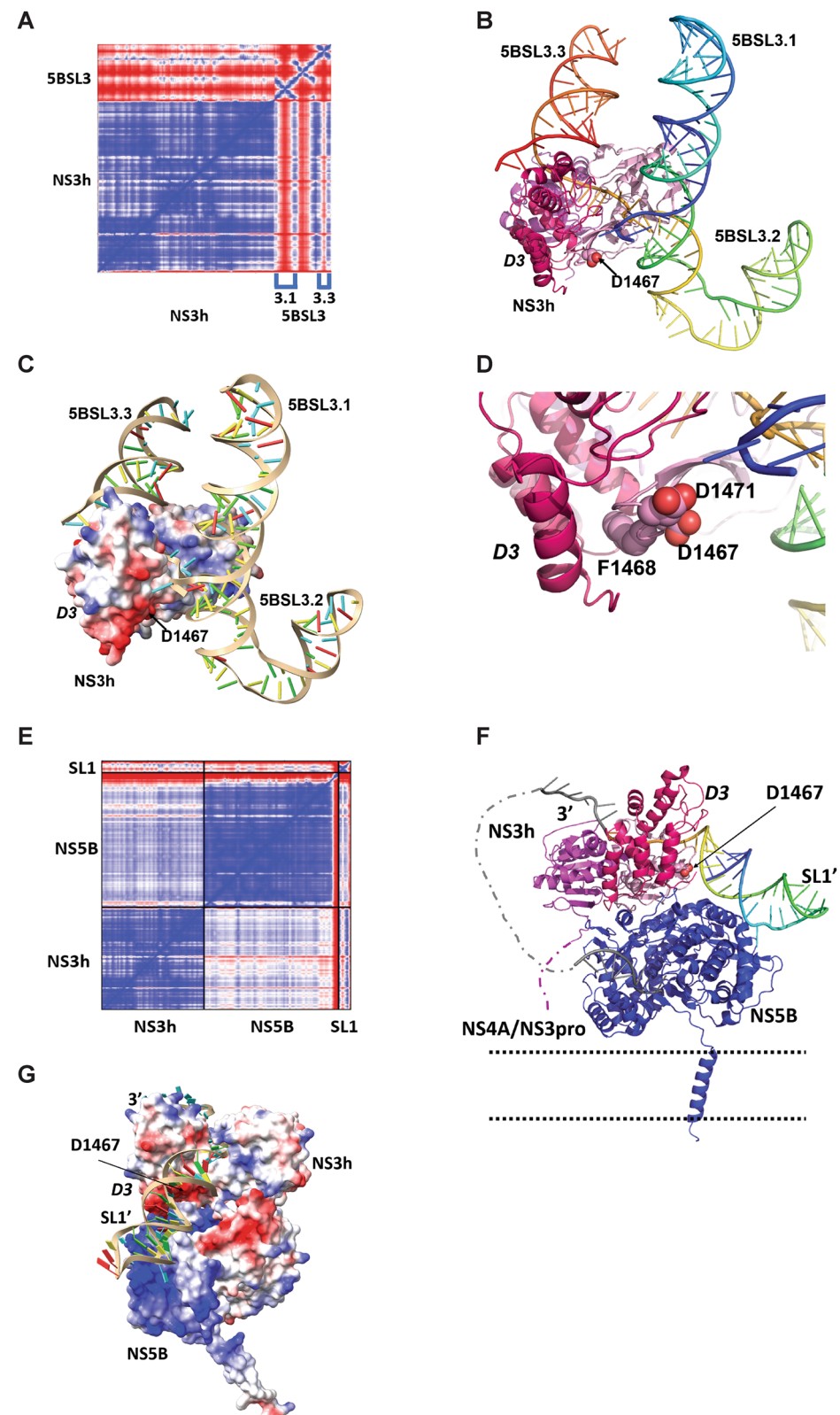

PWPI blasticidin TLR3 or empty control vector was used for stable expression of TLR3[32]. HCV replicon constructs previously published and used in this study include: pFK i389 Luc NS3-3' JFH1 ΔGDD[36], pFK i389 Luc NS3-3' JFH1, pFK i389 neo NS3-3' JFH1[34], pFK i341 PiLuc NS3-3' GLT1E[39], pFK i341 PiLuc NS3-3' JFH1 (PI wt)[105], pFK i341 PiLuc NS3-3' JFH1/5'Con1[46], pFK i341 PiLuc NS3-3' JFH1/3'Con1[46]. For infection experiments, pFK i389 JcR2A[38], pFK i389 JcR2A ΔGDD, pFK i389 JcR2A

ΔE1E2 were used. For protein expression, pTM NS-3' JFH1[103], pTM NS3/4A JFH1[103] and pET24 NS3 2a JFH1 wt[106] were used.

Several constructs were generated in this study using the primers specified in Supplementary Table 1 or restriction digest: pFK i389 Luc NS3-3' JFH1 D1467G, pFK i389 neo NS3-3' JFH1 D1467G, pFK i341 PiLuc NS3-3' GLT1E D1463G, pFK i389 Luc NS3-3' JFH1 G2U, pFK i389 Luc NS3-3' JFH1 A1655G, pFK JcR2A D1467G, pTM NS3-3' JFH1 D1467G, pTM NS3/

**Fig. 9 | AlphaFold predictions of a complex between NS3h and 5BSL3 (A–D) and of a putative NS3h–NS5B–RNA priming complex (E–G) for synthesis of the (−) strand. A** Predicted aligned error (PAE) plot for a complex between the helicase domain NS3h and the cis-acting element of the (+) strand, 5BSL3. Colour code is from red (large PAE) to blue (low PAE, i.e., pairs of residues in correct relative positions). Note that the low PAE values between NS3h and the 5BSL3 cluster in two pairs of streaks correspond to 5BSL3.1 and 5BSL3.3. **B** This prediction is displayed in cartoon representation, with NS3h coloured magenta (domain 1), pink (D2) and dark pink (D3), and 5BSL3 in rainbow colours from blue (5' end) to red (3' end). D3 is labelled in italics on all structural panels. D1467 is displayed as spheres and labelled. A cif file containing the AF3 prediction is included in the Supplementary Data 3. **C** The same complex where NS3h is depicted as an electrostatic surface. **D** Zoom in on the Phe-loop. F1468, D1467, and D1471 are displayed as spheres and labelled.

**E** PAE plot of the best prediction obtained for a complex between the helicase domain NS3h, the polymerase NS5B and the 3'-end of the (+) strand HCV RNA, here a variant of stem-loop 1 (SL1'). **F** This NS3h–NS5B–SL1' complex displayed as in **B**, with NS5B blue and SL1' in rainbow colours. For the latter, extra bases have been added and are depicted in grey to picture how the 3' end of the (+) strand would come out of NS3h and are connected by the dotted grey line to the template-binding groove of NS5B. For comparison of this model to the recently published structures of dengue virus complexes, see Supplementary Fig. 8. The parallel dotted black lines denote the membrane in which NS3 is anchored through its interaction with NS4A and NS5B through its transmembrane C-terminal helix. A cif file containing the AF3 prediction is included in the Supplementary Data 3. **G** The same complex as in **F** seen from the other side with NS3h and NS5B depicted as electrostatic surfaces.

4A JFH1 D1467G, pET16 NS3/4A JFH1 D1467G, pET24 NS3 2a JFH1 D1467G. pET16 NS3/4A JFH1 was obtained by inserting the gene encoding NS3-4A into the multiple cloning site of pET16 to generate an N-terminally His-tagged protein.

### Production of lentiviral particles
Lentiviral particles were generated in HEK293T cells by poly-ethylenimine (PEI) based transfection. $5 \times 10^6$ HEK 293T cells were seeded in 10 cm dishes in DMEM complete. The next day, the seeding media was removed and replaced with 5 mL DMEM (without additives) 30 min prior to transfection. 5.1 μg pWPI Empty or TLR3, 5.1 μg pSPAX2-Gag-pol and 1.7 μg pMD2-VSVG[107] were mixed with 800 μL Opti-MEM and 36 μL PEI and incubated for 20 min at RT. Afterwards, the transfection mix was added dropwise to HEK293T cells. Media was changed to DMEM complete after 4–6 h. After 48 h, lentiviral supernatants were harvested, 0.45 μm filtered and stored at −80 °C.

### TLR3-based directed evolution approach
Huh7 neoJFH1 cells[34] were reversely transduced with lentiviral particles encoding either TLR3 or empty control. Cells were cultured for 8–10 weeks in DMEM complete medium containing 1 mg/mL G418. In the first experiment, emerging single clone colonies of TLR3 transduced cells were expanded. Since empty transduced cells did not form colonies, bulk cells were analysed. To increase comparability in the second experiment, cells of both conditions were seeded in a 15 cm dish at low density to allow single clone colony formation. Single clones were expanded, and RNA was isolated using the Nucleospin RNA extraction kit (Macherey Nagel). Reverse transcription was performed using the Super Script IV First Strand Synthesis System (Thermo Fisher Scientific) using the primers specified in Supplementary Table 1. HCV replicase was amplified using Phusion Flash PCR (Thermo Fisher Scientific). Sanger sequencing was performed using primers covering the whole HCV replicase (Supplementary Table 1) and the Economy Run Tube service at Microsynth seqlab. Sequences were aligned to the reference sequence using the Benchling software.

### In vitro transcription (ivt) and electroporation of HCV RNA
Ivt was performed as described previously[35]. Plasmid DNA was linearized using the respective restriction enzyme (JFH1: MluI, GLT1E: SpeI) (NEB) and purified. Reaction buffer was added at a final concentration of 80 mM HEPES, pH 7.5; 12 mM $MgCl_2$, 2 mM Spermidine, 40 mM DTT. NTPs were added to a final concentration of 3.125 mM. 100U RNase inhibitor (Promega) was added, and ivt was initiated by the addition of 6 μL (90 U) T7 polymerase. Ivt was performed overnight at 37 °C. Ivt RNAs were purified by phenol−chloroform purification. The integrity of the purified ivt RNA was analysed by agarose gel electrophoresis.

Huh7-Lunet empty/TLR3 or Huh7 MAVS KO MAVScr were electroporated with purified ivt RNA as described previously[35]. Cells were detached, counted and resuspended in electroporation buffer (120 mM KCl, 0.15 mM $CaCl_2$, 10 mM $K_2HPO_4$/$KH_2PO_4$ (pH 7.6), 25 mM HEPES, 2 mM EGTA, 5 mM $MgCl_2$, 2 mM ATP, 5 mM glutathione, pH 7.6)

at $1*10^7$/mL. $2 \times 10^6$ cells were electroporated with 5 μg RNA in 0.2 cm cuvettes at 166 mV, 975 μF using the Gene Pulser II (Bio-Rad). Cells were diluted in DMEM complete and seeded at the desired density.

### Analysis of replication using luciferase reporter
Replication was monitored by luciferase reporter expression at the indicated time points as described previously[35]. A 4-h time point was included as a transfection control. In brief, cells were washed once with PBS and lysed by the addition of luciferase lysis buffer (1% Triton X-100, 25 mM Glycyl-Glycin (pH 7.8), 15 mM $MgSO_4$, 4 mM EGTA, 1 mM DTT). Lysed cells were frozen at −20 °C for at least two hours. Luciferase expression was analysed by addition of luciferase substrate in luciferase assay buffer (0.25 M Glycyl-Glyin, 0.015 M $KPO_4$, pH 7.8, 0.015 M $MgSO_4$, 4 mM EGTA, pH 7.8, 1 mM DTT, 2 mM ATP, 0.07 mM Luciferin) using the Mithras LB 940 plate luminometer (Berthold Technologies). Luciferase signals were quenched by the addition of 10% SDS before analysis of the next well.

### Analysis of TLR3 activation
TLR3 activation was analysed either by electroporation of Huh7-Lunet TLR3 cells with ivt RNA of subgenomic reporter replicons or by transducing Huh7-Lunet neoJFH1 wt/D1467G with TLR3 expressing lentivirus. In the first case, cells were electroporated as described above and seeded in 12-well plates. After 72 or 96 h, cells were lysed, and RNA was isolated using the RNA Nucleospin RNA extraction kit (Macherey Nagel). cDNA was generated using a high-capacity cDNA synthesis kit (Thermo Fisher Scientific). IFIT1 and GAPDH were detected using iTaq Universal SYBR Green supermix (Bio-Rad) and specific primers (Supplementary Table 1). Fold change was calculated using the $2^{-\Delta\Delta cT}$ method[108].

In the latter case, Huh7-Lunet neoJFH1 wt/D1467G were reversely transduced with TLR3 expressing lentivirus by co-seeding of cells with respective lentivirus and 4 μg/mL polybrene in 12-well plates. The media was changed after 24 h, and cells were harvested at 24, 48 and 72 h. RNA isolation, cDNA generation and quantification of IFIT1 and GAPDH were performed as described above[32] using specific primers and probes (Supplementary Table 1). HCV RNA was detected by Taqman qPCR using qScript XLT One-Step RT-qPCR kit (Quanta Biosciences).

### Immunofluorescence
Huh7-Lunet naïve, Huh7-Lunet neoJFH1 wt and Huh7-Lunet neoJFH1 D1467G were seeded in 96-well glass-bottom plates (Cellvis). The next day, cells were fixed with 50 μL 4% PFA. Cells were permeabilized with 0.1% Triton X-100 and blocked with blocking buffer (5% normal goat serum in PBS). Afterwards, primary antibodies (Supplementary Table 2) diluted in blocking solution were added and incubated in a humid chamber at 4 °C overnight. Primary antibodies were detected using Alexa fluorophore-labelled isotype-specific secondary antibodies (Thermo Fisher Scientific). Cellular DNA was stained using Hoechst 33342. Images were acquired using the Celldiscoverer 7

(Zeiss). Images were segmented using Cellpose software, and cellular intensity was measured using FIJI.

## DsRNA bioassay
$2 \times 10^6$ Huh7-Lunet naïve or neoJFH1 wt/D1467G were seeded in a 10 cm dish and harvested 24 h after seeding, or Huh7-Lunet empty cells electroporated with indicated constructs were harvested 72 h post-transfection. RNA was isolated using RNAzol (Sigma Aldrich) following the manufacturer's protocol. DsRNA levels in 5 µg total RNA were analysed using the Lumit dsRNA detection assay (Promega) following the manufacturer's protocol. Luminescence was detected using a Mithas LB 940 plate luminometer (Berthold Technologies) and normalized on Huh7-Lunet naïve or ΔGDD control.

## Northern blotting
Huh7-Lunet cells were electroporated with the indicated constructs, lysed after 72 h using RNAzol (Sigma Aldrich), and RNA was isolated following the manufacturer's protocol. To analyse assay sensitivity, (+) and (−) RNA standards were generated by ivt of pFK i389 Luc NS3-3' JFH1 wt using T7 and T3 polymerase, respectively. The (+) strand standard was quantified by spectroscopy and diluted to indicated concentrations. The (−) strand standard could not be quantified due to the presence of multiple transcription products of varying sizes. Both standards were mixed with RNA isolated from Huh7-Lunet naïve. HCV (+) and (−) RNA were detected by Northern Blot, similar to the protocol described previously[46]. RNA was denatured by mixing with 100 mM $NaPO_4$, pH 7, 6 M glyoxal (pH > 5) and DMSO and incubating for 1 h at 50 °C. Samples were mixed with glyoxal loading buffer (0.25 mg/mL bromophenol blue, 0.25 mg/mL xylene cyanol, 10 mM $NaPO_4$, pH 7, 50% glycerol) and run on a 1% agarose gel containing 10 mM $NaPO_4$ pH 7 with 10 mM $NaPO_4$, pH 7, running buffer. RNA was vacuum blotted onto a positively charged nylon membrane (Merck). The gel was covered with 50 mM NaOH during blotting. The membrane was briefly dried and UV cross-linked. To verify RNA integrity and transfer efficiency, membranes were stained with 0.2% methylene blue, 0.3 M sodium acetate pH 5.5, leading to prominent bands of ribosomal RNAs. The membrane was cut below the large ribosomal RNA and destained in distilled water. Digoxigenin-labelled probes were generated by ivt, including DIG-labelled Uracil using pFK i389 Luc NS3-3' JFH1 wt as template, generating 9 or 7 kb DIG-labelled HCV replicon transcripts of complementary polarity for (−) and (+) HCV RNA probes, respectively. T7 polymerase (Roche) or T3 polymerase (Promega) was used to transcribe probes detecting (−) and (+) strands, respectively. ß-Actin probe was generated using pBSK ß-Actin and T7 polymerase (Roche). Probes were denatured for 5 min at 95 °C before use. The upper half of the cut membrane was incubated with HCV probes, while the lower half was incubated with a ß-Actin specific probe. Hybridization was carried out in Easy Hyb hybridization solution (Sigma-Aldrich) overnight at 68 °C. To remove excess probe material, membranes were washed with 2× SSC, 0.1% SDS for 10 min at 68 °C, 0.2× SSC, 0.1% SDS for 10 min at 68 °C and 0.1× SSC, 0.1% SDS for 10 min at 80 °C. Bound probe was detected using anti-Digoxigenin-AP Fab fragments (Roche). Northern blots were imaged using CSPD AP substrate (Roche) and the WB imager (Intas Science Imaging). Band intensity was quantified using ImageJ.

## In vitro helicase unwinding and ATPase assay
The truncated C-terminally His-tagged NS3 proteins lacking the protease domain (NS3h) were expressed and purified as described by Hanson et al.[109].

Molecular beacon-based helicase assay was performed as described previously[109] using the substrate shown in Fig. 6A. To estimate initial velocities of unwinding, readings obtained after ATP injection were fit to a first-order rate equation. The velocity was calculated by multiplying the fitted rate constant by the fitted reaction amplitude.

The relative affinity of the protein for the helicase substrate ($K_{0.5}$) and maximal unwinding velocity ($V_{max}$) were estimated by fitting initial velocities to protein concentration using a Michaelis–Menten equation. Titrations with each protein were performed in triplicate, and the averages are reported. Uncertainties are standard deviations.

ATP hydrolysis was monitored using the procedure described previously[110] with the following modifications. Reactions were performed in clear 96-well microplates, initiated by injecting 5 µL of ATP into 20 µL of a solution containing helicase and nucleic acid such that final concentrations were 25 mM MOPS, 2 mM $MgCl_2$, 5 nM NS3h, 0.01 mg/ml BSA, 0.07% Tween20. After 10 min at 23 °C, reactions were terminated by adding 100 µL of the malachite green reagent (3 volumes 0.045% (w/v) malachite green: 1 volume 4.2% ammonium molybdate in 4 N HCl: 0.05 volume 20% Tween 20). $A_{630}$ was read, and the net phosphate produced was calculated from a phosphate standard curve after subtracting values obtained in control reactions lacking helicase. Specific activities (moles ATP cleaved/mole NS3h/s) are fit to Eq. (1), with constants as defined by the scheme in Fig. 6D. Sequences of DNA/RNA oligonucleotides are located in the supplementary materials (Supplementary Table 1). The pppRNA oligonucleotide was transcribed from a corresponding duplex DNA sequence using NTPs and T7 RNA polymerase.

$$V/E = k_{fast}*[NA]/K_{NA} + [NA] + k_{slow} \tag{1}$$

## AlphaFold predictions
The protein sequences of JFH1 NS3, NS3h and NS5B and RNA sequences from the JFH1 3'X (the last 98 nucleotides in the 3'UTR) or from 5BSL3 (nucleotides 9281-9423) were used as input for AlphaFold3 (AF3)[67] through the beta AF3 server at https://golgi.sandbox.google.com/. Results were downloaded locally and visualized with Pymol (The PyMOL Molecular Graphics System, Version 3.0, Schrödinger, LLC). Electrostatic potential surfaces were computed and visualized with ChimeraX[111] using the "coulombic" command. Predicted aligned errors (PAE) and other statistics were plotted using an in-house Python notebook for Google Colab available at https://colab.research.google.com/github/tubiana/practicals_AI-biology-genetics/blob/main/Display_AF3_results.ipynb. Comparisons with the Dengue virus RNA replicase priming complex (Protein Data Bank (PDB) 8GZP) and elongation complex (PDB 8GZR) were made after manual superimposition of the central sheets of the palm subdomains.

## Sequence alignment
2554 HCV complete polyprotein sequences were retrieved from HCV GLUE[112] and aligned using ClustalOmega through the msa package[113]. Sequence logos were generated with the ggseqlogo package[114].

## Bacterial expression of NS3/4A
To express NS3/4A in Bacteria, BL21(DE3) cells were electroporated with pET16 NS3/4A JFH1 wt/D1467G and plated on Carbenicillin-containing Agar plates. Colonies were picked and cultivated in 1 L LB-medium containing carbenicillin. Expression of NS3/4A was induced by the addition of 0.25 mM IPTG, cold shock, and incubation at 25 °C for 4 h. Bacterial cells were pelleted and lysed using a microfluidizer. NS3/4A was purified by Ni-NTA chromatography and analysed by SDS–PAGE, Coomassie blue staining and WB.

## Transient expression of HCV non-structural proteins
$5 \times 10^5$ Huh7-Lunet T7 cells were seeded in six-well plates. The next day, cells were transfected with the respective pTM plasmid using Transit-LT1 (Mirus Bio) following the manufacturer's protocol. Therefore, 250 µL serum-free Opti-MEM media, 2.5 µg respective pTM plasmid and 7.5 µL Transit-LT1 were mixed and incubated at RT for 15–30 min. The media was replaced with 1.25 mL fresh DMEM complete, and the

transfection mix was added dropwise to the cells. After 24 h, cells were harvested for Western blotting.

## SDS−polyacrylamide gel electrophoresis (PAGE) and Western blotting (WB)

Huh7-Lunet neoJFH1 or transfected Huh7-Lunet T7 cells seeded in six-well plates were lysed with 100 μL lysis buffer (50 mM Tris−HCl pH 7.4, 150 mM NaCl, 1% Triton X-100) supplemented with protease inhibitor (Roche). Nuclei were removed by centrifugation at $18,800 \times g$ for 15 min at 4 °C. Afterwards, the supernatant was transferred to a fresh tube and 6× Lämmli buffer was added. Samples were incubated at 95 °C for 10 min and stored at −20 °C.

Bacterially expressed and purified NS3/4A was adjusted to the same concentration, mixed with 6x Lämmli buffer, incubated at 95 °C for 10 min and stored at −20 °C.

Samples were analysed by SDS−PAGE (10%) and transferred to PVDF (0.45 μm) or nitrocellulose membranes (0.45 μm, only NS5B) by wet blot transfer (400 mA, 1.5 h). Afterwards, membranes were blocked with blocking buffer (5% milk, 0.5% Tween in PBS) and incubated with antibodies in blocking buffer overnight at 4 °C. Primary antibodies were detected by HRP-labelled secondary antibodies (Supplementary Table 2) and described as for NB above.

## NS3 co-Immunoprecipitation

For immunoprecipitation, the cell lysates were harvested as described above for SDS−PAGE and Western blotting, but native protein buffer (NPB; 50 mM Tris−HCl pH 7.5, 150 mM NaCl, 1% NP-40, 1% sodium deoxycholate, 0.1% SDS) was used. NS3 2E3 antibody (Biofront Technologies) was bound to protein G dynabeads (Thermo Fisher Scientific). Of the clarified WCL, 35 μL was mixed with Lämmli buffer and inactivated at 95 °C for 10 min (Input), while 65 μL was added to the beads and incubated overnight at 4 °C (IP). Beads were washed three times and denatured by the addition of Lämmli buffer and incubation at 95 °C for 10 min.

## Analysis of infectious particle production

Huh7-Lunet CD81[high] [102] cells were electroporated with JcR2a wt and D1467G Renilla luciferase reporter constructs. After 72 h, the cell culture supernatant was transferred onto freshly seeded cells and cells were lysed to analyse replication as described previously[38]. 72 h post supernatant transfer, cells were lysed to analyse replication in infected cells.

## Enhanced crosslinking and immunoprecipitation (eCLIP)

Huh7-Lunet neoJFH1 wt and D1467G were seeded in each 6 × 15 cm dish per duplicate and grown until confluent. For UV crosslinking culture medium was removed and cells were washed once with ice-cold PBS, followed by cross-linking (254 nm) at a total dose of 0.8 J/cm$^2$ UV light on ice. Cells were scraped from the tissue culture dish in ice-cold PBS and centrifuged at $400 \times g$ for 5 min at 4 °C. The cell pellet was washed once with ice-cold PBS and then snap-frozen in liquid nitrogen before storage at −80 °C. Pellets were thawed on ice and lysed in 2X lysis buffer (100 mM Tris−HCl pH 7.5, 300 mM NaCl, 2 mM EDTA, 2% (v/v) IGEPAL CA-630, 1% sodium deoxycholate, 0.5 mM TCEP (Tris(2-carboxyethyl)phosphine hydrochloride) with EDTA-free Protease Inhibitor Cocktail (Sigma-Aldrich) for 20 min on ice. An equal volume of nuclease-free H$_2$O was added to fresh lysates, followed by sonication (10% amplitude, 0.7 s on/2.3 s off, 2 kJ). Immunoprecipitation (IP) and cDNA library preparation were performed as described previously[62]. Briefly, unprotected RNA was trimmed by limited RNA digestion with RNase I, followed by IP with a HCV anti-NS3 antibody (HCV anti-NS3 2E3). Per mg of total protein, 5 μg antibody were coupled to 30 μl Protein G beads, up to a maximum of 20 μg antibody and 120 μl Protein G beads per IP. Afterwards, IP and size-matched input (SMI) samples were first separated by SDS−PAGE and then transferred onto a nitrocellulose membrane. The expected size region was extracted, and RNA was released by Proteinase K digestion. IP and SMI RNA were converted into cDNA libraries, which were amplified with 15−20 PCR cycles and finally sequenced using the Illumina NextSeq 500 platform.

## eCLIP data analysis

Paired-end sequencing libraries with read lengths of 2 × 40 nucleotides were adaptor- and quality-trimmed using cutadapt (v1.18). Reads shorter than 18 nt were discarded. A custom Java programme was applied to identify and clip the unique molecular identifier (UMI) associated with each read. The trimmed reads were then aligned to the human (hg38, Ensembl release 110) and the neo-replicon containing all HCV sequences derived from JFH1 using STAR (v2.7.10a)[115] with the parameters --outFilterScoreMinOverLread 0 --outFilterMatchNminOverLread 0 --outFilterMatchNmin 0 --outFilterType Normal --alignSoftClipAtReferenceEnds No --alignSJoverhangMin 8 --alignSJDBoverhangMin 1 --outFilterMismatchNoverLmax 0.04 --scoreDelOpen −1 --alignIntronMin 20 --alignIntronMax 3000 --alignMatesGapMax 3000 --alignEndsType EndToEnd. UMI-aware deduplication was performed with Picard's MarkDuplicates. Protein binding regions were predicted by using MACS2[116], which models read coverage enrichment in an IP sample over a paired SMI control under a Poisson distribution and was used previously for the analysis of eCLIP studies investigating protein binding to viral RNA[62,117–120]. While MACS2 was designed for peak calling on ChIP-seq data, it can be applied to eCLIP data when the shift size modelling is disabled using the --nomodel parameter. Thus, for reads mapped to the positive strand of the replicon, the parameters: --d-min 25 --scale-to small --nomodel --extsize 25 --max-gap 1 --shift 20 were applied, and for reads mapping to the negative strand of the replicon, the parameters --d-min 25 --scale-to small --nomodel --shift -10 --extsize 100 --max-gap 1 --min-length 5 were applied. These parameter setting ensures the calling of peaks at the 5' and 3' ends for both strands. The resulting peaks in wt and D1467G were combined, and overlapping intervals were merged by extending to the respective outermost interval boundary. The identified and merged MACS2 peaks were further filtered by applying a one-sided Fisher's exact test. Statistically significant enrichment was determined by calculating the odds ratio of mapped reads within each peak against all remaining mapped reads between IP and SMI. The Benjamini−Yekutieli[121] procedure was applied to correct for multiple testing, and only peaks with an adjusted $P$ value < 0.05 and a log2-fold change (IP over SMI) greater than two were considered for further analysis. Additionally, the differential binding affinity between D1467G and wt was calculated for all peaks detected in wt and D1467G by performing another Fisher's exact test comparing the read coverages in IP between both conditions, similar to the enrichment analysis between IP and SMI, but without considering an additional log2 fold change threshold.

Enriched crosslinking sites were defined by the first nucleotide in 5'-direction at the 5'-end of read 2 (R2) that overlaps a significantly enriched peak. The coverage of each crosslinking site in IP and SMI was calculated as the number of R2 counts sharing the same 5'-end. Statistical enrichment of each crosslinking site in IP over SMI was performed by applying the same statistical approach used for the identification of significantly enriched peaks.

To visualize the eCLIP signal in IP relative to SMI, the relative information content of IP over SMI[122–124] was calculated as $p_i \times \log_2(p_i/q_i)$, where $i$ represents a genomic position, $p_i$ denotes the relative fraction of aligned reads covering that position in IP and $q_i$ denotes the relative fraction of aligned reads covering the same position in SMI. The relative information content was visualized using the integrative genome visualization (IGV) browser.

## Statistical analysis

A two-sided unpaired $t$-test was performed using GraphPad Prism if not specified otherwise. $P$-values are indicated in the respective figures.

## Reagents, software, and instruments

Detailed information on oligonucleotides, antibodies, reagents, software, and specialized instruments used in this study is provided in Supplementary Tables 1–5.

## Reporting summary

Further information on research design is available in the Nature Portfolio Reporting Summary linked to this article.

## Data availability

Next-generation sequencing data have been deposited to GEO and can be accessed with the accession number GSE292413. The remaining data underlying this article are available in the article and in its online supplementary material. Source data are provided with this paper.

## Code availability

The computer code for the custom analyses is publicly available at https://github.com/AlexGa/cRIP-eCLIP-workflow as of the date of publication.

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

## Acknowledgements

We thank R. Klein and U. Herian for excellent technical assistance. We are grateful to T. Wakita for providing the JFH1 isolate and thank C.M. Rice for generously providing the HCV NS5A 9E10 antibody. We would like to acknowledge the microscopy support from the Infectious Diseases Imaging Platform (IDIP) at the Center for Integrative Infectious Disease Research, Heidelberg, Germany. This work was supported by Deutsche Forschungsgemeinschaft (DFG) [grant numbers 519777725, 272983813 (TRR179)] to V.L. and by ANRS-Maladies infectieuses émergentes [grant numbers ANRS0380 to S.B. and ANRS0548b to T.T.]. Funding for open access charge: Deutsche Forschungsgemeinschaft (DFG) and Heidelberg University. Further support comes from the Helmholtz Association under the Helmholtz Young Investigator Group Programme, [VH-NG-128 to M.M.] and the European Research Council (ERC) [ERC-StG COVIDecode, 101040914 to M.M.].

## Author contributions

P.Ra.: Methodology, conceptualization, formal analysis, visualization, writing, review and editing. S.B.: Formal analysis, methodology, writing, review and editing. L.M.G.: Formal analysis, methodology, writing, review and editing. A.G.: Formal analysis, methodology, writing, review and editing. P.Ro.: Formal analysis, methodology, review and editing. K.J.P.: Formal analysis, methodology, review and editing. T.T.: Formal analysis, methodology, writing, review and editing. M.M: Formal analysis, writing, review and editing. D.N.F: Formal analysis, methodology, writing, review and editing. V.L.: Conceptualization, formal analysis, visualization, writing, review and editing.

## Funding

## Competing interests

The authors declare no competing interests.
