## [Peer Review file · Nature Communications]

Hepatitis C virus NS3 helicase contributes to (-) strand RNA synthesis

Corresponding Author: Professor Volker Lohmann

Version 0:

Reviewer comments:

Reviewer #1

(Remarks to the Author)

The investigators have addressed an important question of significance to the fields of virology and helicase enzymology. The hepatitis C virus is a positive-stranded RNA virus that encodes its own helicase termed non-structural protein 3 (NS3). The helicase activity is required for viral replication, however, the specific function of NS3 is not known. In this manuscript, the investigators devise a genetic screen that selected for a single mutation, D1467G. This point mutation was found to facilitate (-)-strand synthesis. Also, the activation of TLR3 was abrogated in the presence of the D1467G variant. D1467 was found to be completely conserved in 2553 sequences.

The investigators found that the mutation, D1467G, resulted in the same ATPase activity and similar Unwinding activity as the wt enzyme. They introduced the point mutation into an HCV sub-genomic replicon in which they find an impact on the (-) strand synthesis. This is one of the first, if not the first description of a specific function performed by the helicase that assists the replication of viral RNA.

The manuscript is thorough in its application of several advanced methods to identify the RNA binding sites through which NS3 binds to the RNA including eClip. The manuscript will be of interest to a large number of researchers who study RNA viruses.

The manuscript would be improved if the authors provided a more in depth view of the structure by making the Alpha fold model available as a PDB file. If this is not possible, then the description of the NS3 Phe-loop could be significantly improved if the structure of this loop were shown more clearly and from more angles. As is, the NS3 structures are difficult to compare with NS3 as shown in the literature (for example, the structures shown in PNAS from the Rice lab).

Reviewer #2

(Remarks to the Author)

In their manuscript "Hepatitis C virus NS3 helicase contributes to (-) strand RNA synthesis" Ralfs et al. propose a new model in which the hepatitis C virus helicase interacts with the 3' end of the viral genome and the viral RNA-dependent RNA polymerase NS5B to initiate (-) strand RNA synthesis. They expressed TLR3 in Huh-7 Lunet cells to enable an innate immune response suppressing HCV replicon replication and sequenced subgenomic replicon variants that emerged under that pressure. One mutation, D1467G in the NS3 helicase domain, was found consistently across multiple replicates. It reduced viral replication and abolished infectious particle production but most importantly, the mutant showed significantly reduced innate immune activation via TLR3 and RIG-I like receptor signaling which was linked to a reduced abundance of (-) strand HCV and subsequently dsRNA. Although the mutation is located in the NS3 helicase domain, helicase and ATPase function were not altered by D1467G in vitro. Instead, the authors found that D1467G decreased the interaction of NS3 with NS5B. Other subgenomic replicon variants with alterations in the 3'UTR phenocopied the D1467G mutant suggesting NS3 might interact with the 3'UTR. Using eCLIP-seq they found that NS3 D1467G had reduced interaction with a stem loop in the NS5B coding region that had previously been shown to have regulatory functions in HCV replication and translation. Based on their data, which they complement with an AlphaFold 3 prediction, they propose a model in which the helicase NS3 forms complex with the polymerase NS5B and the 3' terminal end of the (+) strand viral genome and unwinds the 3'-terminal stem loop to enable (-) strand priming by the viral polymerase. Overall, this is a very interesting study providing new insights into

the HCV replicative mechanism.

Major comments

1. TLR3 expression in cells used in Fig. 1 should be confirmed by western blot.
2. In general, quantitative data should be shown comprehensively instead of as one representative results (e.g. Fig. 1D, E, G, H but also other figures).
3. The data in Fig. 3E-G shows about 2-fold lower HCV RNA levels, reduced levels of NS3 protein and reduced levels of dsRNA relative to total RNA. These observations are compatible with the D1467G mutant having an about 2-fold reduced replication and do not demonstrate that dsRNA levels are specifically reduced in D1467G mutant replicating cells. Although they do not calculate it, the authors seem to base their conclusion on a reduced dsRNA/NS3 ratio for D1467G replicating cells compared to wt replicating cells (lines 493-497). It is unclear why this ratio would be more meaningful than the ratio of dsRNA/HCV RNA – which seems to be comparable for both replicon variants suggesting no specific reduction in dsRNA but a general reduction in both HCV RNA and dsRNA.
4. If – despite point 3 – the authors wanted to draw conclusions from NS3 protein expression levels, they should show that the antibody detects the mutant just as well as it detects the wild-type protein. This could be done by epitope tagging and ectopically expressing NS3 and showing that the intensity ratios for detection of NS3 and the epitope tag are the same for both variants.
5. For Huh7-Lunet neoJFH1 D1467G cells that stably express the replicon and are cultured over extended periods of time, it should be shown that the mutation does not revert and that no additional mutations arise.

Minor comments

1. Figure 1D is missing the labeling of which cell lines were used for the upper and lower graphs.
2. Descriptions in figure legend 3 do not all refer to the right panels.
3. Presumably cells in Fig. 7E expressed TLR3 – this should be indicated in the figure legend.
4. It would be informative to include a graph about the composition of the GLUE database including the representation of genotypes.

Reviewer #3

(Remarks to the Author)

Reviewer #4

(Remarks to the Author)

In this manuscript Ralfs et al. studied the role HCV helicase NS3 in viral RNA replication, specifically its role in (-) strand RNA synthesis. The authors identified D1467G mutation in the NS3 helicase domain (NS3h) critical for HCV replication. By exploiting this mutation, they show that disrupting NS3h leads to reduced levels of HCV dsRNA. Since (-) strand amount is the limiting factor for HCV dsRNA formation, the authors further investigate the role of NS3 in (-) strand HCV RNA synthesis. Through rich variety and elegant experiments, the authors show that D1467G mutation in NS3h perturbs the NS3 - NS5B protein-protein as well as NS3 - 5BSL3.1 protein-RNA interactions. However, it remains unclear exactly how D1467G mutation affects the (-) strand RNA synthesis.

This work resides outside of my field of expertise, and I was specifically invited by the editor to comment on the eCLIP experiment. Therefore, I will refrain from assessing the significance and contribution of the results to the field and largely focus on the eCLIP experiment.

Major points:

- 1) Choice of MACS2 for eCLIP peak calling is rather unusual given the availability of tools specific for eCLIP or tools that work with all CLIP-seq approaches such as Skipper (<https://github.com/yeolab/skipper>) and RCRUNCH (<https://github.com/zavolanlab/rcrunch>). MACS2 assumes that peak regions are relatively broad and uses a well-defined model for fragment size shift which is not true for RNA-binding proteins. Moreover, MACS2 does not consider Crosslink-Induced Truncation Sites (CITS) for peak calling. Perhaps the choice of peak calling is suitable for identifying viral RBP-RNA interactions? The authors should justify the choice of their eCLIP data analysis approach to clarify this point.
- 2) In the eCLIP experiment, the authors have mapped the reads to human and HCV genome in an appropriate manner. However, it is puzzling why they provide the peaks identified only on the HCV genome. The authors should provide all the peaks identified (including the hg38 peaks) both in GEO submission and in the supplementary table. This could strengthen their findings and rule out the possibility of observed effects are an indirect consequence of potential NS3 binding to stem loop structures present in human transcriptome.
- 3) The authors show that D1467G mutation in NS3h disrupts the interaction with 5BSL3.1 and suggest that this contributes to (-) strand RNA synthesis. To solidify the eCLIP results the authors should provide a second line of independent evidence of NS3 - 5BSL3.1 interaction. EMSA is one of the suitable approaches where the authors can test wt-NS3 and D1467G-NS3 against an RNA construct carrying 5BSL3.1.

4) In order to link the NS3 - 5BSL3.1 interaction to (-) strand HCV RNA synthesis, authors should provide a direct link of evidence where the disruption of 5BSL3.1 specifically results in reduced (-) strand HCV RNA. Similar to luciferase and northern blot experiments that are part of Figure 4, the authors should generate a construct lacking 5BSL3.1 and test its effect on HCV dsRNA levels and +/- RNA ratio. This would clarify whether NS3 - 5BSL3.1 interaction is required for (-) strand synthesis.

5) Why did the authors not consider using a dual-luciferase approach for the luciferase experiments? This would improve the accuracy of the results and rule out the potential biases arising from transfection efficiency and cell survival after transfections.

Minor points:

1) The authors cited an earlier work, Schmidt et al 2023 (PMID: 37794589), referring to the eCLIP methodology. It seems that they used the same snakemake pipeline from that study. If this is the case, providing a link to the GitHub page of the pipeline would improve the reproducibility of this work.

2) In Figure 1D, empty and TLR3 labels are missing for higher and lower panels of luciferase experiments.

3) Line 330 two words "JFH-1" and "using" do not have space in between them.

In conclusion, as a non-expert, it is unclear to me how D1467G mutation is directly linked to (-) strand synthesis. Is it through disrupting the NS3 - NS5B interaction, NS3 - 5BSL3.1 interaction or a combination of both? A major revision incorporating the experiments suggested above is required to sufficiently elucidate the role of NS3 in (-) strand HCV RNA synthesis.

Reviewer #5

(Remarks to the Author)

In the manuscript entitled "Hepatitis C virus NS3 helicase contributes to (-) strand RNA synthesis", Ralfs P. and collaborators identified a mutation (D1467G) in the NS3 helicase domain that impairs synthesis of the negative-strand RNA. They thoroughly characterized the molecular basis of this phenotype and found that the mutation reduces NS3's interaction with both the NS5B RNA polymerase and the 3' end of the positive-strand viral genome, which serves as promoter for genome replication. They proposed a model of the negative-strand RNA synthesis initiation with first the interaction of NS3 with NS5B, followed by the transfer of the protein complex to the 3'NTR, enables NS3 helicase unwinding the terminal stem-loop of the HCV genome rendering the template accessible for de novo initiation of (-) RNA strand synthesis by NS5B.

Overall, the results presented appear to confirm previous observations indicating that interactions between NS3 helicase, NS5A and NS5B are required for the initiation of HCV RNA synthesis (e.g., Binder et al., 2007). Additionally, interactions involving NS3 helicase, NS5B and the 3'X RNA element have been shown to be important for efficient viral replication and particle formation (Murayama et al., 2007).

Specific comments:

- NS3 D1467G mutation was selected under TLR3-dependent selection pressure in the context of an HCV subgenomic replicon. The authors observed massive cell death upon HCV replication in the presence of TLR3 (line 370), which contrasted with the survival observed with the HCV replicon harboring the mutation D1467G in NS3 helicase domain. They showed that replication of the mutant strain is reduced, resulting in lower intracellular levels of dsRNA, presumably below the TLR3 detection threshold.

To further quantify the impact of TLR3 on HCV RNA replication harboring the NS3 mutation, the authors used a HCV replicon with luciferase as readout. Under this condition, they were able to detect viral replication of wild-type HCV genome (Fig. 1D) despite previously reporting (Fig. 1A) massive cell death in the presence of TLR3. Can the authors clarify the differences between the 2 viral replicon systems and explain how wild-type HCV replication was detected despite the previously reported TLR3-mediated degradation?

In addition, Figure 1D lacks an indication of which conditions received TLR3 and which received the control (empty vector).

- Figure 2E: It lacks RNA replication levels of HCV gt1b without TLR3

- Figure 3B and 3C: I am wondering why RLRs pathway is activated in the case of HCV replication, given that viral RNAs produced during infection are 5'-capped with flavin adenine dinucleotide molecule.

- Figure 3F and 3G: It would have been helpful if viral dsRNA quantification had been performed in an absolute manner (e.g., copy number), rather than using a protein-based assay such as the split-nanoluciferase assay.

- Lines 606-624 : The choice to study in vitro helicase activity with only the NS3 helicase domain, rather than the NS3 full-length protein, may limit the interpretation of the data, given that the NS3 protease domain is required for RNA unwinding by NS3 (Beran et al., 2007).

- Figure 7F and 7G: To investigate the impact of the NS3 D1467G mutation on the reduction of HCV replication, the authors assessed protein-protein interactions between NS3 wt or NS3 mutant with other viral replicases (NS3, NS4B, NS4A and NS5B) using co-immunoprecipitation. However, the results are not entirely convincing: interactions between NS3/NS4A and NS3/NS5A are barely detectable and not quantified, while the well-established NS3/NS5B interaction appears weak. The use of an additional method (e.g., pull-down, proximity ligation assay) could strengthened these findings.

- The authors subsequently studied the binding of NS3 D1467G mutant to viral (+) and (-) strand RNAs using the eCLIP approach, and compared the footprinting results to those obtained with the wild-type NS3. While this is a cutting-edge technique, it provides only a qualitative snapshot at a given time point. To strengthen the conclusions, the authors should consider adding a quantitative analysis (such filter-binding assay) to determine whether the D1467G mutation alters the binding affinity of NS3 for viral RNAs. This additional analysis would support the conclusion of the reduced binding of NS3 to (-) SL (line 737).

- Regarding the section of the MS that used AlphaFold3 :

To strengthen the reliability of the AF3 prediction, it would be helpful to present the electrostatic surface complementarity between NS3 and NS5B and their interaction with RNA.

Line 735 "AlphaFold modelling": modelling is not appropriate. Structural models are produced based on experimental data; AF produces predicted structures.

Line 735 "highlights a possible role": since it is a possible role "suggests" is more appropriate than "highlights"

Lines 750-751: "The best AF3 models were obtained with...", then "in the best model as judged by PAE", what about the other best models?

Lines 742 "We got sporadic but definite hits"; line 759 "At any rate": these phrasings are confusing. It is hard to understand how an AF prediction can be "sporadic and definite". I simply don't understand the use of 'at any rate' to introduce the hypothesis "our NS3hel-SL1'-NS5B model may represent a 'priming complex'".

Version 1:

Reviewer comments:

Reviewer #2

(Remarks to the Author)

The authors have adequately addressed all the points I had raised during the prior round of review. As stated before this is an exciting study providing new insights into the function of the the HCV NS3 helicase during viral genome replication.

Reviewer #3

(Remarks to the Author)

Reviewer #4

(Remarks to the Author)

I thank the authors for their thorough revision, which addresses all of my major and minor suggestions. The revised manuscript is significantly improved.

Specifically, the revised manuscript:

- 1- Provides a clearer explanation of the eCLIP methodology and data analysis, with improved reproducibility.
- 2- Includes additional tables detailing the NS3 binding profile, which are essential for the proper interpretation of the results and enhance the manuscript's overall value to the field.
- 3- Presents new experiments and analyses that further explore NS3 binding to SL1 and 5BSL3.
- 4- Offers revised interpretations and conclusions of the eCLIP data. Together with the new AlphaFold predictions, the manuscript now presents a more coherent narrative on the role of NS3 in (-) strand RNA synthesis.

I have one minor comment: please check line 358 — I believe "MAVS" should be "MACS2".

Overall, I support the publication of this manuscript following the revision.

Reviewer #5

(Remarks to the Author)

All issues and suggestions have been considered and addressed in the revised manuscript

Point-by-point response to the reviewers' comments

Reviewer #1:

The investigators have addressed an important question of significance to the fields of virology and helicase enzymology. The hepatitis C virus is a positive-stranded RNA virus that encodes its own helicase termed non-structural protein 3 (NS3). The helicase activity is required for viral replication, however, the specific function of NS3 is not known. In this manuscript, the investigators devise a genetic screen that selected for a single mutation, D1467G. This point mutation was found to facilitate (-)-strand synthesis. Also, the activation of TLR3 was abrogated in the presence of the D1467G variant. D1467 was found to be completely conserved in 2553 sequences.

The investigators found that the mutation, D1467G, resulted in the same ATPase activity and similar Unwinding activity as the wt enzyme. They introduced the point mutation into an HCV sub-genomic replicon in which they find an impact on the (-) strand synthesis. This is one of the first, if not the first description of a specific function performed by the helicase that assists the replication of viral RNA.

The manuscript is thorough in its application of several advanced methods to identify the RNA binding sites through which NS3 binds to the RNA including eClip. The manuscript will be of interest to a large number of researchers who study RNA viruses.

We are grateful for the overall very positive general comments to our manuscript!

1. The manuscript would be improved if the authors provided a more in depth view of the structure by making the Alpha fold model available as a PDB file. If this is not possible, then the description of the NS3 Phe-loop could be significantly improved if the structure of this loop were shown more clearly and from more angles. As is, the NS3 structures are difficult to compare with NS3 as shown in the literature (for example, the structures shown in PNAS from the Rice lab).

We thank the reviewer for bringing up this point. We have added a zip archive containing the AF3 predictions and pymol scripts enabling visualization of the AF3 models to the Supplementary data (Supplementary dataset 2). The predictions now include two complexes, the new one of NS3h with cis-acting RNA 5BSL3 to match the eCLIP data. Accordingly, we have expanded Fig. 9 with now five structure panels including a zoom on the Phe-loop.

Reviewer #2:

In their manuscript “Hepatitis C virus NS3 helicase contributes to (-) strand RNA synthesis” Ralfs et al. propose a new model in which the hepatitis C virus helicase interacts with the 3' end of the viral genome and the viral RNA-dependent RNA polymerase NS5B to initiate (-) strand RNA synthesis. They expressed TLR3 in Huh-7 Lunet cells to enable an innate immune response suppressing HCV replicon replication and sequenced subgenomic replicon variants that emerged under that pressure. One mutation, D1467G in the NS3 helicase domain, was found consistently across multiple replicates. It reduced viral replication and abolished infectious particle production but most importantly, the mutant showed significantly reduced innate immune activation via TLR3 and RIG-I like receptor signaling which was linked to a reduced abundance of (-) strand HCV and subsequently dsRNA. Although the mutation is located in the NS3 helicase domain, helicase and ATPase function were not altered by D1467G in vitro. Instead, the authors found that D1467G decreased the interaction of NS3 with NS5B. Other subgenomic replicon variants with alterations in the 3'UTR phenocopied the D1467G mutant suggesting NS3 might interact with the 3'UTR. Using eCLIP-seq they found that NS3 D1467G had reduced interaction with a stem loop in the NS5B coding region that had previously been shown to have regulatory functions in HCV replication and translation. Based on their data, which they complement with an AlphaFold 3 prediction, they propose a model in which the helicase NS3 forms complex with the polymerase NS5B and the 3' terminal end of the (+) strand viral genome and unwinds the 3'-terminal stem loop to enable (-) strand priming by the viral polymerase. Overall, this is a very interesting study providing new insights into the HCV replicative mechanism.

We thank this reviewer for highlighting the importance of our study.

Major comments

1. TLR3 expression in cells used in Fig. 1 should be confirmed by western blot.

We have analysed TLR3 expression in Huh7 Lunet empty and TLR3 cells by Western blot and added the data to the new Supplementary Figure 2.

2. In general, quantitative data should be shown comprehensively instead of as one representative results (e.g. Fig. 1D, E, G, H but also other figures).

We agree with the reviewer and have replaced all respective figures by comprehensive graphs that display information about all replicates of the underlying data.

3. The data in Fig. 3E-G shows about 2-fold lower HCV RNA levels, reduced levels of NS3 protein and reduced levels of dsRNA relative to total RNA. These observations are compatible with the D1467G mutant having an about 2-fold reduced replication and do not demonstrate that dsRNA levels are specifically reduced in D1467G mutant replicating cells. Although they do not calculate it, the authors seem to base their conclusion on a reduced dsRNA/NS3 ratio for D1467G replicating cells compared to wt replicating cells (lines 493-497). It is unclear why this ratio would be more meaningful than the ratio of dsRNA/HCV RNA – which seems to be comparable for both replicon variants suggesting no specific reduction in dsRNA but a general reduction in both HCV RNA and dsRNA.

We have included NS3 staining as an in-assay control that was analysed side-by-side with the dsRNA staining and see it as a complementation to HCV RNA levels. However, we agree with the reviewer that limited conclusions can be drawn from the NS3 IF due to the differences in background staining intensity, protein half-life etc. Following the suggestion in point 2 of this reviewer, we now show the mean values of the independent replicates, which makes the data easier to interpret. In addition, we have rephrased the respective sentences, highlighting the mentioned limitations of the analysis.

4. If – despite point 3 – the authors wanted to draw conclusions from NS3 protein expression levels, they should show that the antibody detects the mutant just as well as it detects the wild-type protein. This could be

done by epitope tagging and ectopically expressing NS3 and showing that the intensity ratios for detection of NS3 and the epitope tag are the same for both variants.

Since we agree with the reviewer on the limitations of this experiment, we have toned down the respective comments in the results section, see also response to specific point 3.

5. For Huh7-Lunet neoJFH1 D1467G cells that stably express the replicon and are cultured over extended periods of time, it should be shown that the mutation does not revert and that no additional mutations arise.

This is indeed a plausible scenario, which we envisaged in our experimental design. During our initial TLR3-based directed evolution approach, we have identified the single nucleotide change A4741G (GAC→GGC) leading to an amino acid change from aspartate to glycine (D1467G). When designing the neoJFH1 D1467G replicon construct we introduced two mutations at the nucleotide level (A4741G C4742G), resulting in a GGG codon, still encoding a glycine but requiring two mutations for reversion to aspartate. We have sequenced stable cell lines after 6 weeks of culturing and did neither detect reversion of D1467G nor any other dominant mutations. We have included the data in the new Supplementary Figure 3 and explained the strategy in the result section to Figure 1.

Minor comments

1. Figure 1D is missing the labeling of which cell lines were used for the upper and lower graphs.

We apologise for this mistake and have corrected it.

2. Descriptions in figure legend 3 do not all refer to the right panels.

We apologise for this mistake and have corrected it.

3. Presumably cells in Fig. 7E expressed TLR3 – this should be indicated in the figure legend.

We thank the reviewer for bringing up this point. We have indicated the cell line in the figure legend.

4. It would be informative to include a graph about the composition of the GLUE database including the representation of genotypes.

We agree with the reviewer that this is important supporting information. We have added a pie chart illustrating the composition of the curated data base at the time point of usage in the new Supplementary Figure 4.

Reviewer #3:

We thank this reviewer for the contribution to the peer review of our manuscript.

Reviewer #4:

In this manuscript Ralfs et al. studied the role HCV helicase NS3 in viral RNA replication, specifically its role in (-) strand RNA synthesis. The authors identified D1467G mutation in the NS3 helicase domain (NS3h) critical for HCV replication. By exploiting this mutation, they show that disrupting NS3h leads to reduced levels of HCV dsRNA. Since (-) strand amount is the limiting factor for HCV dsRNA formation, the authors further investigate the role of NS3 in (-) strand HCV RNA synthesis. Through rich variety and elegant experiments, the authors show that D1467G mutation in NS3h perturbs the NS3 - NS5B protein-protein as well as NS3 - 5BSL3.1 protein-RNA interactions. However, it remains unclear exactly how D1467G mutation affects the (-) strand RNA synthesis.

This work resides outside of my field of expertise, and I was specifically invited by the editor to comment on the eCLIP experiment. Therefore, I will refrain from assessing the significance and contribution of the results to the field and largely focus on the eCLIP experiment.

We thank this reviewer especially for providing the specific expertise for reviewing the eCLIP data.

Major points:

1) Choice of MACS2 for eCLIP peak calling is rather unusual given the availability of tools specific for eCLIP or tools that work with all CLIP-seq approaches such as Skipper (<https://github.com/yeolab/skipper>) and RCRUNCH (<https://github.com/zavolanlab/rcrunch>). MACS2 assumes that peak regions are relatively broad and uses a well-defined model for fragment size shift which is not true for RNA-binding proteins. Moreover, MACS2 does not consider Crosslink-Induced Truncation Sites (CITS) for peak calling. Perhaps the choice of peak calling is suitable for identifying viral RBP-RNA interactions? The authors should justify the choice of their eCLIP data analysis approach to clarify this point.

We agree with the reviewer and would like to clarify how MACS2 was used in our analysis (also see lines 337 onwards). Briefly, we do not use the shifting model mentioned by the reviewer. Instead, we use MACS2 parameters that produce peak calls highly consistent with the observed eCLIP signal, as represented by the relative information content in IP versus size-matched input samples¹⁻³. When comparing multiple peak calling algorithms, MACS2 preserves the visually observed binding pattern very well, potentially better than more complex peak callers. For this reason, we routinely display MACS2 peak calls together with the relative information in IP versus size-matched input, as shown in Figure 8. The use of MACS2 for the analysis of eCLIP data, particularly on viral RNA, is now well documented⁴⁻⁸. While MACS2 does not consider crosslink-induced truncations, we extract this information and compare crosslinking patterns in IP versus size-matched input samples as described above and displayed in Figure 8. We have rephrased the respective Methods section to improve clarity.

2) In the eCLIP experiment, the authors have mapped the reads to human and HCV genome in an appropriate manner. However, it is puzzling why they provide the peaks identified only on the HCV genome. The authors should provide all the peaks identified (including the hg38 peaks) both in GEO submission and in the supplementary table. This could strengthen their findings and rule out the possibility of observed effects are an indirect consequence of potential NS3 binding to stem loop structures present in human transcriptome.

We agree with the reviewer and have included the requested table detailing any significantly enriched peaks (adjusted $P < 0.05$, \log_2 fold change > 2) detected in the human genome (Supplementary Table 7). We also provide .bed files for viewing the identified peaks in any genome browser with the accompanying GEO submission. Overall, binding of wt and mutant to cellular RNAs is quite similar. We have furthermore explained the data in the Results section of the revised manuscript.

3) The authors show that D1467G mutation in NS3h disrupts the interaction with 5BSL3.1 and suggest that this contributes to (-) strand RNA synthesis. To solidify the eCLIP results the authors should provide a second line of independent evidence of NS3 - 5BSL3.1 interaction. EMSA is one of the suitable approaches where the authors can test wt-NS3 and D1467G-NS3 against an RNA construct carrying 5BSL3.1.

We agree with the reviewer and have revisited the data presentation. In fact, NS3 wt and D1467G both display significant binding to 5BSL3.1 when compared to their matched input controls. However, we defined an enrichment criterion (\log_2 fold change > 2) in addition to statistical significance that was used for peak filtering. The mutant narrowly missed this inclusion criterion (\log_2 fold change: 1.99). Given the relatively small difference in enrichment between wt (\log_2 fold change: 2.5) and mutant, we feel that it is unlikely that this difference is of biological relevance. In the revised version of our manuscript, we have now also included peaks in Supplementary Table 6 that did not meet our enrichment criterion, and we have toned down all associated statements.

Furthermore, we analysed binding of NS3h wt and D1467G to SL1, the likely initiation site of (-) strand synthesis, using different RNA- and DNA-oligonucleotides and a fluorescence polarization based assay as published previously⁹. Here, we did not detect any differences between wt and D1467G in this assay. In general, NS3h bound far better to DNA than to RNA, particularly binding to dsRNA was barely detectable, overall questioning the physiological relevance of in vitro binding assays in the context of viral RNA synthesis. We have added these data to Supplementary Figure 5 and the accompanying text to the Results section.

We further have analysed the interaction of NS3h to 5BSL3 using AF3, yielding a prediction with low predicted alignment error. According to this data, NS3h binds to the single-stranded region between 5BSL3.3 and 5BSL3.2. We have added this data to the updated Figure 9A-D.

4) In order to link the NS3 - 5BSL3.1 interaction to (-) strand HCV RNA synthesis, authors should provide a direct link of evidence where the disruption of 5BSL3.1 specifically results in reduced (-) strand HCV RNA. Similar to luciferase and northern blot experiments that are part of Figure 4, the authors should generate a construct lacking 5BSL3.1 and test its effect on HCV dsRNA levels and -/+ RNA ratio. This would clarify whether NS3 - 5BSL3.1 interaction is required for (-) strand synthesis.

We thank the reviewer for bringing up this point. Deletion of 5BSL3.1 is not possible since it is part of the NS5B coding region and deletion would therefore lead to RNA-structure independent replication defects. However, the significance of 5BSL3.1 for HCV replication has been analysed previously for gt1b Con1^{10,11}. Synonymous mutations disrupting 5BSL3.1 did not affect HCV replication, indicating that 5BSL3.1 does not majorly contribute to negative strand synthesis. Mutations in 5BSL3.3 mildly affected viral replication, while interfering with the structure of 5BSL3.2 or changes in the sequence of the loop region completely abrogated HCV replication, highlighting the functional significance of 5BSL3.2

and its kissing loop-interaction with 3'X for viral replication. Still, our AF3 model indicates binding of NS3h to 5BSL3. We have discussed the rather auxiliary function of this binding in the light of the published genetic data in the corresponding part of the results section.

5) Why did the authors not consider using a dual-luciferase approach for the luciferase experiments? This would improve the accuracy of the results and rule out the potential biases arising from transfection efficiency and cell survival after transfections.

Luciferase-based HCV replicons have been established in the field since their development¹². Electroporation of viral RNA ensures rapid, efficient and consistent delivery of viral RNA and transfection efficiency is analysed by measuring translation of transfected constructs by assessing luciferase activity 4h post electroporation. Usage of this well-established cell culture system ensures high comparability with previously published literature. We have added an explanatory sentence highlighting the role of the 4-hour time point as transfection efficiency control in the Methods section.

Minor points:

1) The authors cited an earlier work, Schmidt et al 2023 (PMID: 37794589), referring to the eCLIP methodology. It seems that they used the same snakemake pipeline from that study. If this is the case, providing a link to the GitHub page of the pipeline would improve the reproducibility of this work.

We agree with the reviewer and are providing the requested GitHub link (<https://github.com/AlexGa/cRIP-eCLIP-workflow>) with the revised version of our paper.

2) In Figure 1D, empty and TLR3 labels are missing for higher and lower panels of luciferase experiments.

We apologise for this mistake and have corrected it.

3) Line 330 two words “JFH-1” and “using” do not have space in between them.

We apologise for this mistake and have corrected it.

In conclusion, as a non-expert, it is unclear to me how D1467G mutation is directly linked to (-) strand synthesis. Is it through disrupting the NS3 - NS5B interaction, NS3 - 5BSL3.1 interaction or a combination of both? A major revision incorporating the experiments suggested above is required to sufficiently elucidate the role of NS3 in (-) strand HCV RNA synthesis.

We hope that we could make the involvement of the HCV helicase and the impact of the mutation clearer by addressing the specific points above.

Reviewer #5 (Remarks to the Author):

In the manuscript entitled “Hepatitis C virus NS3 helicase contributes to (-) strand RNA synthesis”, Ralfs P. and collaborators identified a mutation (D1467G) in the NS3 helicase domain that impairs synthesis of the negative-strand RNA. They thoroughly characterized the molecular basis of this phenotype and found that the mutation reduces NS3's interaction with both the NS5B RNA polymerase and the 3' end of the positive-strand viral genome, which serves as promoter for genome replication. They proposed a model of the negative-strand RNA synthesis initiation with first the interaction of NS3 with NS5B, followed by the transfer of the protein complex to the 3'NTR, enables NS3 helicase unwinding the terminal stem-loop of the HCV genome rendering the template accessible for de novo initiation of (-) RNA strand synthesis by NS5B. Overall, the results presented appear to confirm previous observations indicating that interactions between

NS3 helicase, NS5A and NS5B are required for the initiation of HCV RNA synthesis (e.g., Binder et al., 2007). Additionally, interactions involving NS3 helicase, NS5B and the 3'X RNA element have been shown to be important for efficient viral replication and particle formation (Murayama et al., 2007).

We thank the reviewer for summarizing the key findings of our manuscript. We agree that previous studies, which we have all cited and discussed, indeed pointed to a cooperative role of the helicase in viral RNA replication, in conjunction with NS5B and NS5A. However, these older studies could not at all identify specific functions of the helicase in this process. Our recent paper goes far beyond in providing direct experimental evidence for a specific function of the helicase in the initiation of negative strand synthesis.

Specific comments:

- NS3 D1467G mutation was selected under TLR3-dependent selection pressure in the context of an HCV subgenomic replicon. The authors observed massive cell death upon HCV replication in the presence of TLR3 (line 370), which contrasted with the survival observed with the HCV replicon harboring the mutation D1467G in NS3 helicase domain. They showed that replication of the mutant strain is reduced, resulting in lower intracellular levels of dsRNA, presumably below the TLR3 detection threshold. To further quantify the impact of TLR3 on HCV RNA replication harboring the NS3 mutation, the authors used a HCV replicon with luciferase as readout. Under this condition, they were able to detect viral replication of wild-type HCV genome (Fig. 1D) despite previously reporting (Fig. 1A) massive cell death in the presence of TLR3. Can the authors clarify the differences between the 2 viral replicon systems and explain how wild-type HCV replication was detected despite the previously reported TLR3-mediated degradation?

We thank the reviewer for bringing up this comment. In Figure 1A, we use stable neoJFH1 replicon cell lines which are transduced with TLR3 encoding lentiviruses (comparable to Figure 1G+H), leading to stable TLR3 expression, and passaged for an extended period of time (>6 weeks) in the presence of G418 (HCV selection) and Blasticidin (TLR3 selection). After multiple weeks, we observed massive cell death, likely due to the loss of G418 resistance mediated by HCV clearance from the cells due to the induction of a robust ISG response as in Figure 1H. So, in essence, cell death is induced by G418 treatment, not by HCV replication. In Figure 1D, Huh7 Lunet cells stably expressing TLR3 were electroporated with JFH1 wt/D1467G subgenomic RNA encoding a luciferase reporter gene (shown in Fig. 1C) and replication and IFIT1 expression was analysed up to 96h post electroporation. Here, ISGs are induced in TLR3 expressing but not empty control cells after the onset of replication, likely leading to the observed reduction of JFH1 wt replication in Huh7 Lunet TLR3 but not empty control cells. However, since no cytotoxic drugs are added here, no cell death will occur. To improve clarity, we have added an explanatory sentence to the Results section, highlighting the continuous G418 selection during the screen.

In addition, Figure 1D lacks an indication of which conditions received TLR3 and which received the control (empty vector).

We apologise for this mistake and have corrected it.

- Figure 2E: It lacks RNA replication levels of HCV gt1b without TLR3

We have included the data in the respective figure.

- Figure 3B and 3C: I am wondering why RLRs pathway is activated in the case of HCV replication, given that viral RNAs produced during infection are 5'-capped with flavin adenine dinucleotide molecule.

Activation of RLRs (RIG-I and Mda5) by HCV dsRNA is well established and has been extensively studied¹³⁻¹⁷. While 5'ppp is a main trigger of RIG-I activation¹⁸ especially for short

dsRNA, longer dsRNAs activate RIG-I without the need for 5'ppp¹⁹. In addition, homopolymeric regions within the HCV genome have been reported to further contribute to RIG-I activation^{16,17}. Mda5 is furthermore activated by HCV dsRNA irrespective of 5'ppp^{14,15}. Since RLR signalling is efficiently inhibited by NS3/4A-mediated MAVS cleavage²⁰⁻²² we used cells expressing NS3/4A cleavage resistant MAVS (MAVScr), which have been reported previously to induce ISG expression upon HCV replication¹³.

FAD capping for HCV has been demonstrated only recently²³ and it has been reported that FAD capping reduces RIG-I activation. However, the authors reported reduced, yet significant RIG-I activation in certain experiments (²³ Fig. 4e) and the genomic RNA of the major HCV genotype 1 is not FAD capped at all. Furthermore, the authors used either single stranded oligos, replication-deficient replicon or analysed ISG activation at an early time point (18h) after infection, where viral replication did not yet take place according to the authors and RIG-I was likely activated by incoming ssRNA.

We have added a sentence to the results section, to highlight the detection of HCV by RLRs despite FAD capping.

- Figure 3F and 3G: It would have been helpful if viral dsRNA quantification had been performed in an absolute manner (e.g., copy number), rather than using a protein-based assay such as the split-nanoluciferase assay.

We thank the reviewer for bringing up this point. Absolute quantification of dsRNA is technically challenging since most PCR-based assays detect positive and negative strand separately and generally high positive strand to negative strand ratios impede accurate negative strand detection. In addition, qPCR-based methods do not provide information on whether positive and negative strand RNA are present as single or double stranded RNA. To further investigate the level of positive and negative strand RNA, we performed Northern blot and detected a reduction in negative strand synthesis by D1467G. Northern blot data as shown in Fig. 4 are the most reliable specific quantification method for positive and negative strand RNA, again with the caveat of not being able to differentiate ssRNA from dsRNA. DsRNA-specific antibodies are often used for the detection of dsRNA. We used the J2 dsRNA-antibody and detected that D1467G reduced dsRNA levels close to the levels in naïve cells. To verify these results using a separate, independent method, we applied the split-nanoluciferase based assay, which relies on split-nanoluciferase fragments fused to separate RNA binding domains. Using this assay, we detected a reduction of dsRNA levels in D1467G cells close to levels in naïve cells and therefore confirmed our findings observed with the dsRNA-antibody assay. This data is in our view an important complementation to the quantitative ssRNA detection shown in Fig. 4.

- Lines 606-624: The choice to study in vitro helicase activity with only the NS3 helicase domain, rather than the NS3 full-length protein, may limit the interpretation of the data, given that the NS3 protease domain is required for RNA unwinding by NS3 (Beran et al., 2007).

The NS3 helicase domain has been extensively studied in enzymatic, structural and functional assays and is able to unwind dsRNA or dsDNA in the absence of the protease domain^{9,24-28}. Therefore, the protease domain is not required for unwinding activity and NS3h alone is sufficient to analyse the impact of our mutation on unwinding activity as published previously. Addition of the protease domain increases unwinding efficiency which has been proposed to be due to the formation of a positively charged patch on the back side of the protein, since this effect can be mimicked by addition of an N-terminal His-tag on NS3h^{29,30}. To improve clarity, we have added an explanatory sentence referencing the multiple studies using the helicase domain for in vitro helicase activity in the Results section.

- Figure 7F and 7G: To investigate the impact of the NS3 D1467G mutation on the reduction of HCV replication, the authors assessed protein-protein interactions between NS3 wt or NS3 mutant with other viral replicases (NS3, NS4B, NS4A and NS5B) using co-immunoprecipitation. However, the results are not entirely convincing: interactions between NS3/NS4A and NS3/NS5A are barely detectable and not quantified, while the well-established NS3/NS5B interaction appears weak. The use of an additional method (e.g., pull-down, proximity ligation assay) could strengthen these findings.

We agree that the NS3-NS5B interaction is detectable only to a limited extent, however, robustly and reproducibly. Importantly, we found a significantly reduced interaction for the D1467G mutant. The limited degree of interaction is likely in part due to its transient nature and due to the fact that not all NS3 molecules will interact with NS5B and vice versa at a given timepoint.

Indeed, we so far failed to identify NS3-NS4B or NS3-NS5A interaction. We hypothesized that this is due to the transient nature of these interactions. Therefore, in response to the concerns of this reviewer, we treated replicon cells with the cell permeable, reversible crosslinker DSP prior to harvesting. Under these conditions, we were able to detect interaction of NS3 with NS4B, NS5A and NS5B, indicating that all non-structural proteins are interacting to various degrees. We have included this data in the new Supplementary Figure 7. Importantly, upon crosslinking, we found no difference in the interaction of NS3 wt and NS3 D1467G with NS5B, further highlighting that the mutation reduces either the strength or the duration of the interaction between the proteins.

Further assays assessing interaction of NS3 and NS5B are either not feasible due to lack of specific antibodies (proximity ligation assay) or would require the introduction of artificial tags within the proteins (pull-down) and therefore, even if successful, likely not lead to a deeper mechanistic understanding.

We have addressed this point by adding the new data with the crosslinker (Supplementary Figure 7) and by explaining the conclusions mentioned above in the results section.

- The authors subsequently studied the binding of NS3 D1467G mutant to viral (+) and (-) strand RNAs using the eCLIP approach, and compared the footprinting results to those obtained with the wild-type NS3. While this is a cutting-edge technique, it provides only a qualitative snapshot at a given time point. To strengthen the conclusions, the authors should consider adding a quantitative analysis (such as filter-binding assay) to determine whether the D1467G mutation alters the binding affinity of NS3 for viral RNAs. This additional analysis would support the conclusion of the reduced binding of NS3 to (-) SL α (line 737).

We so far did not claim any differential binding of NS3 to (-) SL α , as indicated by the reviewers' comment, but found a slight difference in binding to 5BSL3.1. However, NS3 wt and D1467G in fact bound to 5BSL3.1 with similar significance. The mutant only slightly missed the second inclusion criterion for significant hits with a \log_2 fold change of 1.99 (criterion: \log_2 FC>2; wt: \log_2 FC=2.5). This makes it very unlikely, that the apparent difference shown in the Figure has a biological relevance. We have now also included the nonsignificant hits in Supplementary Table 6 and toned down all associated statements.

We further have analysed the interaction of NS3h to 5BSL3 using AF3, yielding a prediction with low predicted alignment error. According to this data, NS3h binds to the single stranded region between 5BSL3.3 and 5BSL3.2. We have added this data to the updated Figure 9A-D. Furthermore, we analysed binding of NS3h wt and D1467G to SL1, the likely initiation site of (-) strand synthesis, using different RNA- and DNA-oligonucleotides and a fluorescence polarization based assay as published previously⁹. Here, we did not detect any differences between wt and D1467G. In general, NS3h bound far better to DNA than to RNA, particularly binding to dsRNA was barely detectable, overall questioning the physiological relevance of in

vitro binding assays in the context of viral RNA synthesis. We have added these data to Supplementary Figure 5 and the accompanying text to the Results section.

- Regarding the section of the MS that used AlphaFold3:

To strengthen the reliability of the AF3 prediction, it would be helpful to present the electrostatic surface complementarity between NS3 and NS5B and their interaction with RNA.

We agree with the reviewer that this represents a valuable addition. We have added the illustration as electrostatic surface to Figure 9C+G and point out the electrostatic complementarity between NS3h and NS5B in the results.

Line 735 "AlphaFold modelling": modelling is not appropriate. Structural models are produced based on experimental data; AF produces predicted structures.

We thank the reviewer for bringing up this point. We have rephrased the respective sections.

Line 735 "highlights a possible role": since it is a possible role "suggests" is more appropriate than "highlights"

We have rephrased the respective sentence according to the reviewer's suggestion.

Lines 750-751: "The best AF3 models were obtained with...", then "in the best model as judged by PAE", what about the other best models?

We apologize for the confusion and have rephrased the respective sentence.

Lines 742 "We got sporadic but definite hits"; line 759 "At any rate": these phrasings are confusing. It is hard to understand how an AF prediction can be "sporadic and definite". I simply don't understand the use of 'at any rate' to introduce the hypothesis "our NS3hel-SL1'-NS5B model may represent a 'priming complex'".

We apologize for the confusion and have rephrased the respective section of the Results.

References

- 1 Van Nostrand, E. L. *et al.* A large-scale binding and functional map of human RNA-binding proteins. *Nature* **583**, 711-719 (2020). <https://doi.org/10.1038/s41586-020-2077-3>
- 2 Van Nostrand, E. L. *et al.* Principles of RNA processing from analysis of enhanced CLIP maps for 150 RNA binding proteins. *Genome Biol* **21**, 90 (2020). <https://doi.org/10.1186/s13059-020-01982-9>
- 3 Yee, B. A., Pratt, G. A., Graveley, B. R., Van Nostrand, E. L. & Yeo, G. W. RBP-Maps enables robust generation of splicing regulatory maps. *RNA* **25**, 193-204 (2019). <https://doi.org/10.1261/rna.069237.118>
- 4 Schmidt, N. *et al.* SND1 binds SARS-CoV-2 negative-sense RNA and promotes viral RNA synthesis through NSP9. *Cell* **186**, 4834-4850 e4823 (2023). <https://doi.org/10.1016/j.cell.2023.09.002>
- 5 Basak, A. *et al.* Control of human hemoglobin switching by LIN28B-mediated regulation of BCL11A translation. *Nat Genet* **52**, 138-145 (2020). <https://doi.org/10.1038/s41588-019-0568-7>
- 6 Gonzalez-Perez, A. C. *et al.* The Zinc Finger Antiviral Protein ZAP Restricts Human Cytomegalovirus and Selectively Binds and Destabilizes Viral UL4/UL5 Transcripts. *mBio* **12** (2021). <https://doi.org/10.1128/mBio.02683-20>
- 7 Munschauer, M. *et al.* The NORAD lncRNA assembles a topoisomerase complex critical for genome stability. *Nature* **561**, 132-136 (2018). <https://doi.org/10.1038/s41586-018-0453-z>
- 8 Schmidt, N. *et al.* The SARS-CoV-2 RNA-protein interactome in infected human cells. *Nat Microbiol* **6**, 339-353 (2021). <https://doi.org/10.1038/s41564-020-00846-z>
- 9 Mukherjee, S. *et al.* Identification and analysis of hepatitis C virus NS3 helicase inhibitors using nucleic acid binding assays. *Nucleic Acids Res* **40**, 8607-8621 (2012). <https://doi.org/10.1093/nar/gks623>
- 10 Friebe, P., Boudet, J., Simorre, J. P. & Bartenschlager, R. Kissing-loop interaction in the 3' end of the hepatitis C virus genome essential for RNA replication. *J Virol* **79**, 380-392 (2005). <https://doi.org/10.1128/JVI.79.1.380-392.2005>
- 11 You, S., Stump, D. D., Branch, A. D. & Rice, C. M. A cis-acting replication element in the sequence encoding the NS5B RNA-dependent RNA polymerase is required for hepatitis C virus RNA replication. *J Virol* **78**, 1352-1366 (2004). <https://doi.org/10.1128/jvi.78.3.1352-1366.2004>
- 12 Krieger, N., Lohmann, V. & Bartenschlager, R. Enhancement of hepatitis C virus RNA replication by cell culture-adaptive mutations. *J Virol* **75**, 4614-4624 (2001). <https://doi.org/10.1128/JVI.75.10.4614-4624.2001>
- 13 Bender, S. *et al.* Activation of Type I and III Interferon Response by Mitochondrial and Peroxisomal MAVS and Inhibition by Hepatitis C Virus. *PLoS Pathog* **11**, e1005264 (2015). <https://doi.org/10.1371/journal.ppat.1005264>
- 14 Hiet, M. S. *et al.* Control of temporal activation of hepatitis C virus-induced interferon response by domain 2 of nonstructural protein 5A. *J Hepatol* **63**, 829-837 (2015). <https://doi.org/10.1016/j.jhep.2015.04.015>
- 15 Cao, X. *et al.* MDA5 plays a critical role in interferon response during hepatitis C virus infection. *J Hepatol* **62**, 771-778 (2015). <https://doi.org/10.1016/j.jhep.2014.11.007>
- 16 Saito, T., Owen, D. M., Jiang, F., Marcotrigiano, J. & Gale, M., Jr. Innate immunity induced by composition-dependent RIG-I recognition of hepatitis C virus RNA. *Nature* **454**, 523-527 (2008). <https://doi.org/10.1038/nature07106>
- 17 Uzri, D. & Gehrke, L. Nucleotide sequences and modifications that determine RIG-I/RNA binding and signaling activities. *J Virol* **83**, 4174-4184 (2009). <https://doi.org/10.1128/JVI.02449-08>
- 18 Thoresen, D. *et al.* The molecular mechanism of RIG-I activation and signaling. *Immunol Rev* **304**, 154-168 (2021). <https://doi.org/10.1111/imr.13022>

- 19 Binder, M. *et al.* Molecular mechanism of signal perception and integration by the innate immune sensor retinoic acid-inducible gene-I (RIG-I). *J Biol Chem* **286**, 27278-27287 (2011). <https://doi.org/10.1074/jbc.M111.256974>
- 20 Colasanti, O. *et al.* Comparison of HAV and HCV infections in vivo and in vitro reveals distinct patterns of innate immune evasion and activation. *J Hepatol* **79**, 645-656 (2023). <https://doi.org/10.1016/j.jhep.2023.04.023>
- 21 Meylan, E. *et al.* Cardif is an adaptor protein in the RIG-I antiviral pathway and is targeted by hepatitis C virus. *Nature* **437**, 1167-1172 (2005). <https://doi.org/10.1038/nature04193>
- 22 Li, X. D., Sun, L., Seth, R. B., Pineda, G. & Chen, Z. J. Hepatitis C virus protease NS3/4A cleaves mitochondrial antiviral signaling protein off the mitochondria to evade innate immunity. *Proc Natl Acad Sci U S A* **102**, 17717-17722 (2005). <https://doi.org/10.1073/pnas.0508531102>
- 23 Sherwood, A. V. *et al.* Hepatitis C virus RNA is 5'-capped with flavin adenine dinucleotide. *Nature* **619**, 811-818 (2023). <https://doi.org/10.1038/s41586-023-06301-3>
- 24 Tai, C. L., Chi, W. K., Chen, D. S. & Hwang, L. H. The helicase activity associated with hepatitis C virus nonstructural protein 3 (NS3). *J Virol* **70**, 8477-8484 (1996). <https://doi.org/10.1128/JVI.70.12.8477-8484.1996>
- 25 Kim, D. W., Gwack, Y., Han, J. H. & Choe, J. C-terminal domain of the hepatitis C virus NS3 protein contains an RNA helicase activity. *Biochem Biophys Res Commun* **215**, 160-166 (1995). <https://doi.org/10.1006/bbrc.1995.2447>
- 26 Mousseau, G., Kota, S., Takahashi, V., Frick, D. N. & Strosberg, A. D. Dimerization-driven interaction of hepatitis C virus core protein with NS3 helicase. *J Gen Virol* **92**, 101-111 (2011). <https://doi.org/10.1099/vir.0.023325-0>
- 27 Sweeney, N. L. *et al.* Primuline derivatives that mimic RNA to stimulate hepatitis C virus NS3 helicase-catalyzed ATP hydrolysis. *J Biol Chem* **288**, 19949-19957 (2013). <https://doi.org/10.1074/jbc.M113.463166>
- 28 Jennings, T. A. *et al.* NS3 helicase from the hepatitis C virus can function as a monomer or oligomer depending on enzyme and substrate concentrations. *J Biol Chem* **284**, 4806-4814 (2009). <https://doi.org/10.1074/jbc.M805540200>
- 29 Beran, R. K., Serebrov, V. & Pyle, A. M. The serine protease domain of hepatitis C viral NS3 activates RNA helicase activity by promoting the binding of RNA substrate. *J Biol Chem* **282**, 34913-34920 (2007). <https://doi.org/10.1074/jbc.M707165200>
- 30 Frick, D. N., Rypma, R. S., Lam, A. M. & Gu, B. The nonstructural protein 3 protease/helicase requires an intact protease domain to unwind duplex RNA efficiently. *J Biol Chem* **279**, 1269-1280 (2004). <https://doi.org/10.1074/jbc.M310630200>